# Fixing the Broken Compass: Diagnosing and Improving Inference-Time Reward Modeling

## Abstract

Inference-time scaling techniques have shown promise in enhancing the reasoning capabilities of large language models (LLMs). While recent research has primarily focused on training-time optimization, our work highlights inference-time reward model (RM)-based reasoning as a critical yet overlooked avenue. In this paper, we conduct a systematic analysis of RM behavior across downstream reasoning tasks, revealing three key limitations: (1) RM can impair performance on simple questions, (2) its discriminative ability declines with increased sampling, and (3) high search diversity undermines RM performance. To address these issues, we propose **CRISP** (Clustered Reward Integration with Stepwise Prefixing), a novel inference-time algorithm that clusters generated reasoning paths by final answers, aggregates reward signals at the cluster level, and adaptively updates prefix prompts to guide generation. Experimental results demonstrate that CRISP significantly enhances LLM reasoning performance, achieving up to **5%** accuracy improvement over other RM-based inference methods and an average of **10%** gain over advanced reasoning models.

## 1 Introduction

The remarkable achievements of OpenAI's o1 have sparked a wave of research into inference-time scaling techniques in reasoning tasks [21, 6, 42]. Some works aim to enhance models during the training phase, employing reinforcement learning (RL) [38, 23] or supervised fine-tuning (SFT) [41, 19] on high-quality data to equip models with the ability to generate long chains of thought (CoT). Other approaches focus on inference-time optimization, using reward model (RM)-based search strategies such as Monte Carlo Tree Search (MCTS) to guide the model toward more efficient solution paths [35, 29, 43].

Driven by the great success of the DeepSeek-R1 series [6], recent efforts have predominantly focused on reproducing its performance from a training-centric perspective [19, 41, 38], while largely overlooking inference optimization methods. Although R1-style works achieve strong performance on tasks such as math reasoning, they have been shown to suffer from serious issues such as overthinking [4, 31] and limited task generalization [44, 47]. These issues, however, can be mitigated through RM-based inference techniques. For example, on the commonsense reasoning task CSQA [33], DeepSeek-R1-7B [6] achieves 64.8 accuracy with an average of 3,613 tokens. In contrast, our RM-based inference method, applied to its base model Qwen2.5-Math-7B [40], reaches a higher accuracy of **72.0** using only **1,100 tokens** on average. Therefore, optimizing inference-time reasoning remains a critical direction, particularly for smaller models.

How can we further improve the reasoning performance of LLMs at inference time? Revisiting R1-style work, one key insight is their identification of the reward hacking issue during RL training, which they address using rule-based reward functions, ultimately improving performances [16, 6, 7].

This raises a natural question: **Can we similarly analyze the issues of the reward model at inference time and mitigate them to enhance the LLM's reasoning ability?**

In this work, we investigate the factors affecting reward model performance at inference time and propose methods to mitigate its limitations. Specifically, we begin by mathematically modeling the RM-based inference process to identify its key influencing factors: the input questions, the number of sampled responses, and the search parameters. Then, we conduct targeted experiments to analyze the impact of each factor on RM performance: **(1) Input question:** We test the performance of BoN and MCTS across different question difficulty levels and demonstrate that RM-based inference significantly impairs performance on simple questions. **(2) Sampling number:** We analyze the RM's discriminative ability under different numbers $n$ and observe that its performance deteriorates as $n$ increases. The statistical analysis attributes this degradation to an inverse long-tail phenomenon, wherein the RM tends to assign higher scores to low-frequency, incorrect responses. **(3) Search parameters:** We focus on parameters controlling search diversity, such as sampling temperature and MCTS tree structure. Our results show that RM performs best under moderate diversity, while excessive diversity undermines reasoning accuracy.

To mitigate the former issues in RM-based inference, we design a novel algorithm called **CRISP** (**C**lustered **R**eward **I**ntegration with **S**tepwise **P**refixing). CRISP operates in an iterative fashion, where each round begins by sampling reasoning paths conditioned on a dynamic prefix set. These paths are then clustered by their final answers, allowing the algorithm to aggregate reward signals at the cluster level and thereby attenuate the RM's tendency to mis-rank rare but incorrect outputs. We further incorporate an early termination mechanism based on cluster cardinality, which enables efficient inference on simple questions and alleviates RM instability in such cases. Finally, high-scoring paths from dominant clusters inform the construction of stepwise prefixes for the next sampling round, enabling tighter control over search diversity by limiting the number of intermediate states explored. We conduct extensive experiments to compare our method with other baselines. The results not only indicate that our method is effective in improving RM-based reasoning abilities, with accuracy gains of up to **5%**, but also validate the soundness of our earlier findings. Moreover, compared to DeepSeek-R1 models of the same scale, our method reduces average token usage by up to **90%**, while achieving an average accuracy improvement of **10%** on non-mathematical tasks.

Our main contributions are as follows: (1) We draw three critical findings based on a systematic analysis of RM behavior during inference: RM degrades performance on simple questions, fails to effectively distinguish low-frequency incorrect samples, and performs suboptimally under excessive search diversity. (2) We propose CRISP, a novel inference-time algorithm that clusters generated reasoning paths by final answers, aggregates reward signals at the cluster level, and adaptively updates prefix prompts to guide generation, effectively mitigating the shortcomings of reward models at inference time. (3) Extensive experiments demonstrate that CRISP consistently outperforms both inference-time and training-time baselines, with accuracy improvements of up to **5%** compared to other RM-based inference methods, and an average of **10%** over R1 models in non-mathematical reasoning tasks.

## 2 Overall Performance of Reward Models in Inference-Time

We first evaluate the overall performance of the reward model in inference time as our preliminary experiments. Here we compare the accuracy of Best-of-N (BoN), which generates multiple responses and selects the best one based on the reward score.

**Experimental Setup**  For the policy model, we select some representative open-source models: Gemma2-9B [24], Llama3.1-8B [25], Qwen2.5-3B and Qwen2.5-14B [39]. For the evaluation of reward models, we consider several advanced works, including two outcome reward models (ORMs)—ArmoRM [34] and Skywork-Llama-3.1-8B [14]—and two process reward models (PRMs)—Shepherd-Mistral-7B-PRM [35] and Skywork-o1-PRM-Qwen-2.5-7B [20]. These models demonstrate commendable performance on related benchmarks (see Appendix A for details). As for the evaluation data, following previous works [30, 3, 22], we select MATH-500 [10, 12], which consists of high-school competition-level math problems. In addition to BoN, we also set two baselines: **SC** and **Oracle**. For the former, we select the major voting answer from $n$ responses. For the latter, we directly recall the existing correct answer from the generated samples, which serves as the performance ceiling.

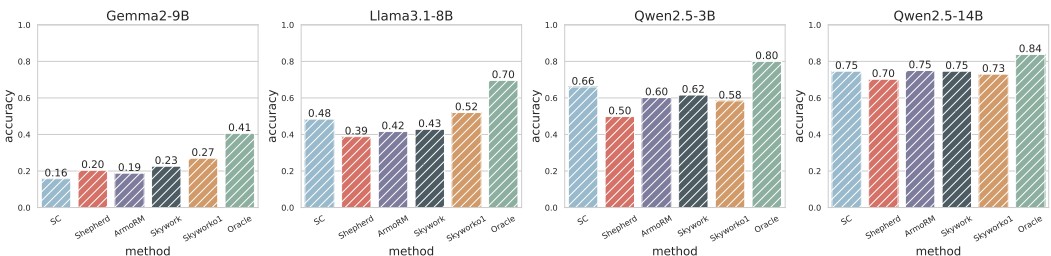

Figure 1: The performance of different policy models using various reward models for BoN inference on the MATH dataset ($n = 10$).

**Main Results** Figure 1 shows the main results of the evaluation (more results, including more datasets and inference strategy in Appendix B). We can conclude that: **Advanced reward models have limited performance on the downstream math reasoning task.** For most LLMs, BoN only provides minor improvements over SC (<5%). Specifically, on Qwen2.5-3B, the BoN for all reward models exhibits lower accuracy than SC, indicating that the BoN inference method has limited reasoning performance. Besides, Oracle significantly outpaces other baselines, suggesting that the performance bottleneck lies in the RM's discriminative ability rather than the LLM's generative capability. Therefore, **identifying and mitigating the factors that impair the RM's performance during inference are crucial for enhancing LLM's reasoning ability**.

## 3 Probing RM-based Inference Issues

### 3.1 Mathematical Modeling

During the inference phase, the first step is to input the question $q$ and generate multiple responses $\mathcal{R}$:

$$\mathcal{R} = \mathcal{S}(\mathcal{M}(q), n; \Phi) \tag{1}$$

where $\mathcal{M}(q)$ denotes the output distribution of the policy model after inputting the question, $n$ denotes the number of samples and $\Phi$ denotes the parameters of the search strategy $\mathcal{S}$ (such as sampling temperature). After that, we use a scoring function $f$ to select the best response $\hat{r}$ from $\mathcal{R}$:

$$\hat{r} = \arg\max_{r \in \mathcal{R}} f(r) \tag{2}$$

To analyze the performance of the reward model, we define $f$ as the score predicted by the RM. Our work focuses on identifying key factors that influence RM performance. To this end, we vary the components in Eq.1 to observe the accuracy of predicted $\hat{r}$ under different $\mathcal{R}$. Specifically, we study three main factors through probing experiments: the input question $q$, the sampling number $n$, and the search parameters $\Phi$.

### 3.2 Experimental Setup

For reward models, based on results in Figure 1, we select the best-performing Skywork and Skywork-o1 as the ORM and PRM for our subsequent experiments. Regarding policy models, we use Qwen2.5-3B and Llama3.1-8B throughout our experiments. To ensure that our findings are not specific to a particular strategy, we conduct all experiments using both BoN and MCTS. As for evaluation data, we employ the MATH-500 dataset in our main text, and provide additional results on GSM8K [5] and OlympiadBench [9] in the appendix.

### 3.3 Input Question: Reward Model Underperforms on Easy Questions

**Question Difficulty Modeling** We first investigate how different questions affect the RM's performance. Following former works, we use question difficulty as a metric to classify different questions [12, 30]. We bin the policy model's pass@1 rate (estimated from 10 samples) on each question into five quantiles, each corresponding to increasing difficulty levels. For example, If the model answers correctly 0 or 1 time, the question is level 5 (hardest). If it answers correctly more than 8 times, the question is level 1 (easiest). Besides, we also study the difficulty approximation without the ground truth and report results in Appendix C.

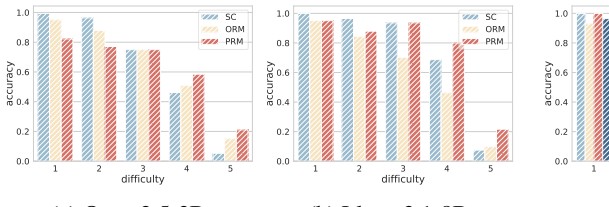

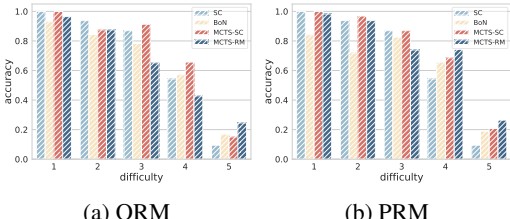

| (a) Qwen2.5-3B | (b) Llama3.1-8B |
|---|---|
| | |
| (a) ORM | (b) PRM |

Figure 2: Performance of BoN inference across different question difficulty levels.

Figure 3: Performance of MCTS inference across different question difficulty levels.

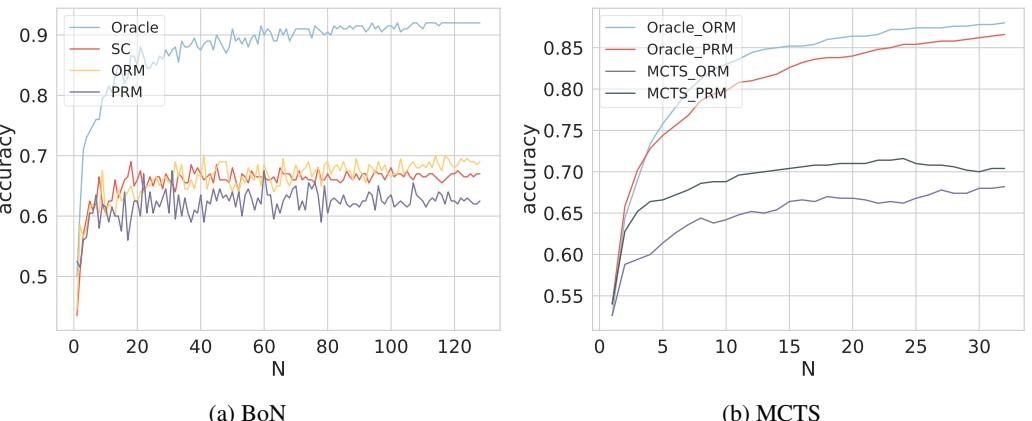

(a) BoN

(b) MCTS

Figure 4: Two inference methods performance across difference sampling numbers.

**BoN Performance**    After categorizing the data by difficulty, we analyze the BoN performance across different levels. We sample 32 examples from each question and illustrate the accuracy in Figure 2, from which we can conclude that: **Compared to SC, BoN performs worse on simple but better on difficult questions.** From the easiest level 1 to the hardest level 5, the accuracy of SC gradually declines, while BoN transitions from lagging behind SC to surpassing it. We also repeat the experiment on two more math reasoning benchmarks and present the results in Appendix D, further confirming our conclusion.

**MCTS Performance**    In MCTS, we use two different scoring functions $f$ to select the final response for comparison: MCTS-SC and MCTS-RM (more functions in Appendix B). For the former, we employ a majority voting method for selection. For the latter, we choose the path with the highest reward score. We perform 32 rollouts over 200 questions, demonstrating the results in Figure 3. Although MCTS provides improvement over BoN, the accuracy of MCTS-RM still lags behind that of SC for low-difficulty problems (see levels 1 and 2 in Figure 3a). Besides, MCTS-SC achieves higher accuracy on easy questions but performs worse on harder questions compared to MCTS-RM. These indicate that: **(Cl.1) The introduction of the RM can hinder the LLM's reasoning performance on simple problems.** This pattern is not limited to specific inference strategies.

### 3.4    Sampling Number: RM struggles to distinguish low-frequency negatives

**Gap between Accuracy and Coverage**    Recent works [3] demonstrate the LLM's coverage of correct answers increases as the sampling number grows, whereas the accuracy does not fully scale with $n$. Based on this, we further investigate whether introducing better RMs and inference strategies can reduce the gap between coverage and accuracy. The changes in accuracy and coverage are shown in Figure 4. The results demonstrate that: **Regardless of the reward model or inference strategy used, the model's accuracy does not improve as $n$ increases.** For both figures, the accuracy plateaus beyond a relatively small number of samples (approximately 30). In contrast, the Oracle setting consistently increases, leading to a persistently widening gap between accuracy and coverage.

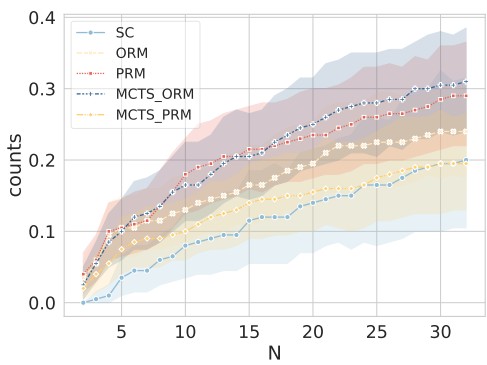
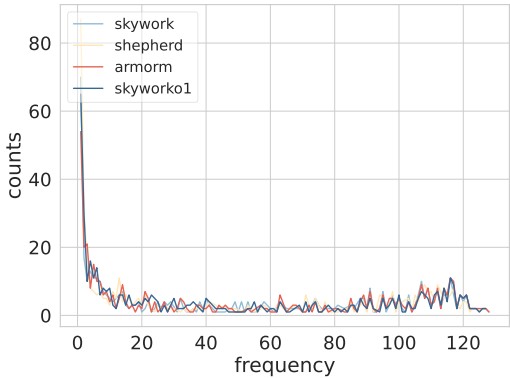

Figure 5: The number of times the model's selection changes from correct to incorrect.

Figure 6: Frequency statistics of the highest-scored negative responses in BoN.

**Discriminative Performance**  In the context of increasing coverage, the policy model's accuracy primarily depends on the reward model's discriminative capacity. Therefore, the plateau observed in Figure 4 is likely due to the reward model selecting incorrect answers as $n$ increases. To validate this claim, we begin with a case study, in which we randomly select a set of questions to examine the correctness of the RM's selections under different sampling numbers (see Appendix E for detailed results). We observe that in some cases, the reward model assigns the highest score to newly generated but incorrect responses, thereby causing originally correct answers to be replaced with incorrect ones as $n$ increases. Additionally, we record the number of instances in which the selected answer transitions from correct to incorrect and present the results in Figure 5. All methods exhibit a tendency for more incorrect transitions as $n$ increases. **This indicates that the model increasingly erroneous distinctions as the sampling size grows**. Moreover, compared to SC, RM-based inference methods show higher transition counts in Figure 5, which suggests that incorporating reward models introduces more incorrect selections.

**Inverse Long-tail Phenomenon**  Why does the reward model perform worse as the sampling number grows? Reflecting on its training process [34, 14, 35], the training data primarily consists of paired responses (i.e., a correct one and an incorrect one). These pairs represent a constrained subset of the response space. We hypothesize that as $n$ grows, more low-frequency responses (those outside the training distribution) are sampled. The reward model struggles to generalize to these unfamiliar inputs, leading to incorrect responses occasionally receiving higher scores. To validate this hypothesis, we perform a statistical analysis of negative responses. For each question, we select the incorrect response with the highest RM score and compute the frequency of its answer across all samples. As shown in Figures 6 and 20, the RM displays an **inverse long-tail phenomenon** when scoring incorrect responses. For most questions, the top-scoring incorrect answers tend to have very low frequencies (frequency < 5 in Figure 6). Conversely, incorrect answers with high occurrence frequencies rarely achieved the highest scores. These findings support our hypothesis: **(Cl.2) RMs struggle to correctly score incorrect responses with low occurrence frequencies, making it difficult to distinguish incorrect responses from correct ones as $n$ grows.**

### 3.5 Search Parameters: RM performs worse on high-diversity distributions

**Search Diversity in BoN**  The final influencing factor we investigate is the search parameters $\Phi$, which are primarily utilized to control the diversity of the policy model's search. For the BoN method, the temperature $T$ is the key parameter controlling the search diversity. We sweep $T$ and analyze its influence on the performance, as shown in Figure 7. For both policy models, BoN performance consistently degrades with increasing $T$, while SC and Oracle (i.e., coverage) remain stable except at high temperatures ($T > 0.9$ in Figure 7). These results indicate that RM is more sensitive to sampling diversity than the policy model. **Higher diversity makes it challenging for the RM to distinguish between positive and negative responses.**  To better understand this issue, we perform additional statistical analyses in Appendix F, which suggest that higher sampling temperatures cause the policy model to produce more low-frequency incorrect responses, thereby degrading discriminative accuracy.

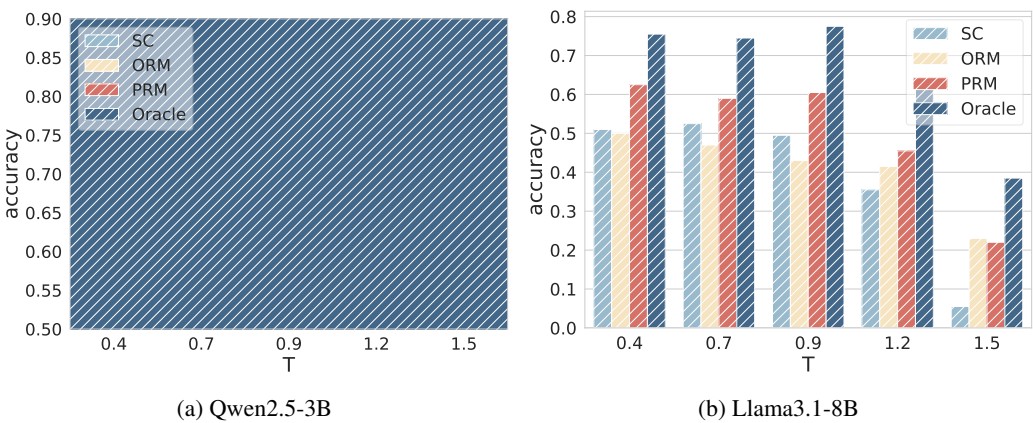

(a) Qwen2.5-3B                         (b) Llama3.1-8B

Figure 7: Performance of BoN inference across different sampling temperatures.

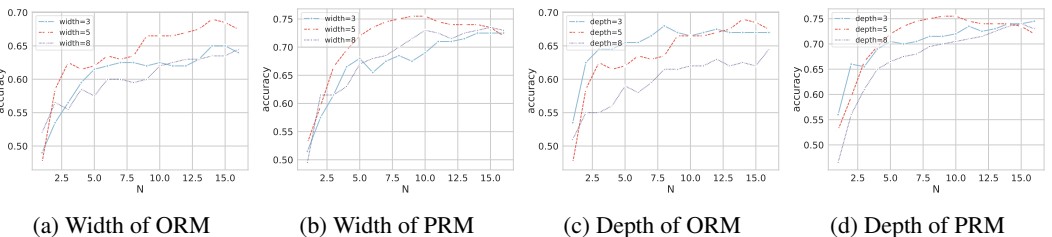

(a) Width of ORM       (b) Width of PRM       (c) Depth of ORM       (d) Depth of PRM

Figure 8: MCTS inference performance under different tree structures.

**Search Diversity in MCTS** In the MCTS algorithm, search diversity is primarily governed by the tree structure, determined by two key parameters: width and depth. The width refers to the number of child nodes at each node, whereas the depth denotes the length of the longest path from the root to a leaf node. A larger width indicates a broader search space during exploration, while a greater depth implies the model can traverse more intermediate states along a single trajectory. We evaluate MCTS performance under varying settings and present the results in Figure 8. The findings reveal: (1) For width, the best performance is observed at intermediate values (width = 5), too high widths lead to a decline in performance. (2) For depth, the best performance is achieved under settings with a lower value (e.g., depth = 3 or 5). These suggest that in MCTS, exploring too many intermediate states can harm performance. Notably, the optimal number of intermediate steps in search does not necessarily align with the number of steps a human would take to solve the same problem. We also analyze the impact of exploration weight on the diversity of MCTS, with consistent findings (see Appendix G). In summary, excessive diversity, such as width, depth, or temperature, can impair the performance of the reward model. Thus, we conclude: **(Cl.3) During inference, it is essential to constrain the diversity of the sampling distribution to maintain the optimal performance of the RM.**

## 4 Mitigating RM-based Inference Issues

### 4.1 Our Methodology

In the preceding sections, we uncover key patterns that affect the RM's performance and identify serval issues in RM-based reasoning. To mitigate these issues, we propose a novel RM-based inference algorithm called **Clustered Reward Integration with Stepwise Prefixing (CRISP)**. Figure 9 and Algorithm 1 demonstrate the main process of our method, which comprises five modules:

**Path Generation** Given a question $q$, during each iteration, we generate new reasoning paths based on the existing prefix set $\mathcal{P}$:

$$\mathcal{R} = \mathcal{R} \cup \mathcal{M}(q, n, \mathcal{P}) \tag{3}$$

In the generation process, the policy model generates $n$ complete sequences of remaining reasoning steps conditioned on $\mathcal{P}$ ($\mathcal{P} = \emptyset$ in the init iteration), rather than generating intermediate nodes step

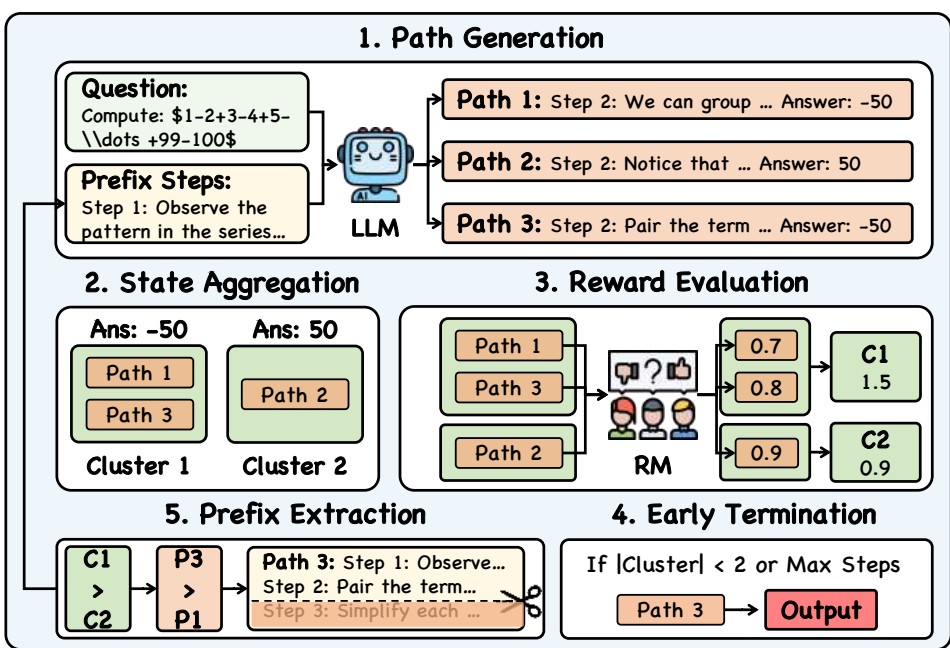

Figure 9: Main process of our CRISP method.

by step as in approaches like MCTS. This helps control the diversity of the search space and reduces the negative impact of excessive diversity on the reward model, as discussed in Cl.3.

**State Aggregation** To further reduce the complexity of the state space and mitigate the impact of low-frequency negative examples on the reward model's performance (as discussed in Cl.2), we define a final-answer-based state aggregation function $\psi$:

$$\psi : \mathcal{R} \to \mathcal{C} \tag{4}$$

where $\mathcal{C}$ is the set of final answer clusters (i.e., all responses leading to the same answer), and for any path $r_1, r_2 \in \mathcal{R}$, we have:

$$\psi(r_1) = \psi(r_2) \iff Answer(r_1) = Answer(r_2) \tag{5}$$

All paths that produce the same final answer are mapped to the same cluster $\mathcal{C}_j \in \mathcal{C}$. As an example, in Module 2 of Figure 9, paths 1 and 3, both with the answer of -50, are assigned to the same cluster.

**Reward Evaluation** After clustering the responses, we can convert the reward scores $f$ for each path into scores $\mathcal{F}$ for the corresponding clusters $\mathcal{C}_j$ (i.e., lines 17-20 in Algorithm 1):

$$\mathcal{F}(\mathcal{C}_j) = \sum_{x \in \mathcal{C}_j} f(x) \tag{6}$$

In the implementation, we normalize $f(x)$ before summing. By additionally considering the frequency of the answers associated with each path during scoring, we can prevent the reward model from assigning excessively high scores to low-frequency responses, thereby mitigating the issue identified in Cl.2. Although this may reduce the scores for some low-frequency correct responses, we will later demonstrate through ablation experiments that this design overall improves performance (see §4.4).

**Early Termination** This module controls when to exit the loop and return the final response. In addition to the standard exit condition of reaching the maximum number of iterations, we also control early termination by monitoring the number of clusters. If the number falls below a certain threshold (set to 2 in our work), it indicates that the question is relatively simple (as evidenced and discussed in Appendix C). In this case, the algorithm terminates, returning the answer corresponding to the most populated cluster, which is equivalent to SC. This not only reduces inference costs but also mitigates the issue of the reward model underperforming on simple questions (see Cl.1).

Table 1: Accuracy comparison in main experiments, the best results are highlighted in **bold**.

| Methods | | Qwen2.5-3B | | | Llama3.1-8B | | |
|---|---|---|---|---|---|---|---|
| | | GSM8K | MATH | Olympiad | GSM8K | MATH | Olympiad |
| CoT | | 0.78 | 0.46 | 0.24 | 0.85 | 0.38 | 0.11 |
| Self-Consistency | | 0.83 | 0.64 | 0.31 | 0.91 | 0.57 | 0.16 |
| Best-of-N | + ORM | 0.83 | 0.65 | 0.31 | 0.91 | 0.47 | 0.18 |
| | + PRM | 0.87 | 0.61 | 0.34 | 0.95 | 0.62 | 0.23 |
| BoN Weighted | + ORM | 0.83 | 0.67 | 0.31 | 0.89 | 0.53 | 0.20 |
| | + PRM | 0.86 | 0.60 | 0.36 | 0.94 | 0.62 | 0.24 |
| MCTS | + ORM | 0.92 | 0.67 | 0.34 | 0.90 | 0.43 | 0.13 |
| | + PRM | 0.95 | 0.71 | 0.31 | 0.95 | 0.57 | 0.19 |
| Beam Search | | 0.95 | 0.73 | 0.34 | 0.94 | 0.56 | 0.15 |
| **Ours** | + ORM | 0.91 | 0.70 | 0.36 | 0.89 | 0.49 | 0.18 |
| | + PRM | **0.96** | **0.76** | **0.39** | **0.95** | **0.67** | **0.26** |

**Prefix Extraction** In this module, we extract the top multiple prefixes as the new prefix set $\mathcal{P}$ for the next iteration, based on the scores of the paths and clusters. As illustrated in Module 5 of Figure 9, we first select the top-$k$ clusters with the highest scores (here, $k$=1, so we select Cluster 1). Then, from the selected cluster(s), we choose the path with the highest score (in this case, $0.8 > 0.7$, so we select Path 3) to extract the prefix. Specifically, at the $i$-th generation, we extract the first $i$ steps of all paths as $\mathcal{P}$, and repeat the process until termination.

## 4.2 Main Experiments

**Experimental Setup** We compare the reasoning performance of our method with other advanced baselines, including: **CoT** [37], **Self-Consistency** [36], **Best-of-N**, **BoN Weighted** [30], **MCTS** [8] and **Beam Search** [30]. For datasets, in addition to MATH-500 [10, 12], we also validate our methods on GSM8K [5] and OlympiadBench [9]. For models, we continue to select Qwen2.5-3B and Llama3.1-8B as the policy model, while using Skywork-Llama-3.1-8B (ORM) and Skywork-o1-PRM-Qwen-2.5-7B (PRM) as the reward model. We present more details in Appendix H.

**Main Results** We demonstrate the result in Table 1, from which we can get the following conclusions: **(1) Our proposed CRISP method significantly improves RM's performance in reasoning tasks.** Across all benchmarks and both model backbones, CRISP consistently outperforms existing RM-based inference approaches. Notably, on the Llama3.1-8B model, CRISP achieves a performance gain of up to **5.0%** on the MATH dataset over the best-competing method. **(2) The findings from the preceding analysis are reasonable.** CRISP is specifically crafted to overcome the key issues of reward modeling revealed in §3. Its consistent and significant performance improvements provide strong empirical evidence that CRISP effectively mitigates these limitations, which are critical bottlenecks affecting the model's reasoning performance.

## 4.3 Training-Time vs. Inference-Time Optimization

To demonstrate the continued necessity of our inference-time optimization approach amid the rising dominance of RL and SFT techniques represented by the DeepSeek-R1 series, we compare our method against the R1 model across different reasoning tasks, including math reasoning (MATH-500), commonsense reasoning (CSQA [33]), social reasoning (SIQA [27]) and logical reasoning (LogiQA [15]). Specifically, given the same base model, we evaluate the accuracy and token consumption among its chat version (using CoT), the R1 distilled version, and our proposed method. From the results in Table 2, we can observe that: **(1) Our method enables more efficient reasoning across all tasks.** It achieves comparable reasoning tokens to the CoT method, while reducing output length by over **90%** compared to the R1 model in the best case. **(2) Our method exhibits stronger generalization capabilities.** Although it underperforms the R1 model on math tasks, it consistently outperforms R1 on other reasoning benchmarks, with average gains of **10%** and **5%** accuracy across two backbones. This highlights the advantage of our inference-time optimization in generalizing across diverse scenarios.

Table 2: Comparison between R1 models and our method, the best accuracy are highlighted in **bold**.

| Base Models | Methods | Math | | Commonsense | | Social | | Logical | |
|---|---|---|---|---|---|---|---|---|---|
| | | Acc | Length | Acc | Length | Acc | Length | Acc | Length |
| Qwen2.5-Math-1.5B | Chat | 0.52 | 1470 | 0.40 | 1400 | 0.46 | 1204 | 0.40 | 2790 |
| | R1-Distill | **0.79** | 13421 | 0.47 | 6066 | 0.52 | 6407 | 0.35 | 12352 |
| | Ours | 0.59 | 943 | **0.58** | 1004 | **0.61** | 1144 | **0.44** | 1143 |
| Qwen2.5-Math-7B | Chat | 0.74 | 1855 | 0.58 | 1479 | 0.58 | 1388 | 0.49 | 2133 |
| | R1-Distill | **0.88** | 9626 | 0.65 | 3612 | 0.66 | 2920 | 0.50 | 6492 |
| | Ours | 0.79 | 987 | **0.72** | 1100 | **0.66** | 1059 | **0.59** | 2058 |

## 4.4 Discussion and Future Work

**Ablation Study**  We perform ablation experiments to validate the contribution of each module in the CRISP framework, with results summarized in Figure 23 of Appendix I. The results show that removing any single module leads to a decline in performance. As our design is informed by the analysis presented in §3 (i.e., Cl.1-Cl.3), the results provide further empirical support for our findings.

**Cost Analysis**  As an inference-time method, in addition to accuracy, reasoning cost is also an important factor to consider. We therefore measure computational cost (e.g., number of generated tokens and inference time) in our evaluations and report the results in Figure 24 of Appendix J. It demonstrates that our CRISP method incurs lower costs compared to other advanced methods.

**Limitations & Future Work**  While our work provides a thorough investigation of RM behavior during inference, it does not address potential issues that may arise during the training of models. In future work, we aim to extend our study to the training phase of reward models. Understanding how training dynamics (such as reward signal design and data sampling strategies) impact downstream reasoning performance could offer deeper insights and help improve the overall reliability of LLM.

## 5 Related Work

**Inference-time Optimization Technique in LLM's Reasoning**  Recent studies have demonstrated that large language models (LLMs) can be effectively enhanced through search-based optimization at inference time [21, 42, 45]. These works primarily follow two approaches: optimizing the strategy for LLMs to search for answers [8, 30, 2, 22] or improving the reward model's ability to evaluate response quality [35, 43, 29]. However, most studies explore these two approaches separately, with limited research analyzing the impact of search factors on RM performance. Our work addresses this gap and proposes a new search strategy to mitigate RM's deficiencies.

**Reward Model in LLM's Reasoning**  The reward model plays a crucial role in complex reasoning tasks of LLMs [42, 29, 35]. Existing works mainly investigate the RM from two perspectives: evaluation and optimization. For the former, researchers design various datasets to evaluate the RM's ability to distinguish between positive and negative responses [11, 17, 46]. For the latter, researchers focus on the training phase, improving the RM's ability by synthesizing high-quality data [35, 14] or optimizing the training algorithm [43, 1, 18]. There is a lack of in-depth analysis of the potential issues RM faces during inference, as well as methods to optimize RM's performance in the inference stage. Our work addresses the gaps left by these related studies.

## 6 Conclusion

In this work, we focus on analyzing key factors that influence the reward model's performance in reasoning tasks. We find that low question difficulty, large sampling number, and high search diversity can lead to issues in RM-based inference, with in-depth explanations provided. To address these issues, we propose CRISP, a cluster-based, prefix-guided inference algorithm that enhances the robustness and efficiency of the reward model. Experimental results demonstrate that our method is effective in enhancing LLM reasoning capabilities.

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

# A Performance of Selected RMs

To demonstrate that the RM issues identified in our experiments in Section §2 are not due to the selected RM's inherently low discriminative abilities, here we present the performance of our RM. For the two ORMs (e.g. ArmoRM-Llama3-8B and Skywork-Reward-Llama-3.1-8B), we report their performance on RewardBench [11] compared to other baselines in Table 3. For the two PRMs (e.g. Math-Shepherd-Mistral-7B-PRM and Skywork-o1-Open-PRM-Qwen-2.5-7B), we report their performance on ProcessBench [11] compared to other baselines in Table 4. From them, we can get that the performance of these models on relevant benchmarks is comparable to the advanced LLMs (e.g. gpt4), hence they are representative.

# B Additional Overall Experiments

In addition to the experiments in the main text, we also conduct the experiments in other settings.

Firstly, while the main text compares different RMs using BoN methods, we now replicate this comparison using the MCTS approach. Our settings are as follows:

- **SC:** Using the self-consistency method for comparison;
- **Reward:** Using the reward score as $f$ in MCTS (e.g. MCTS-Reward in §3.3);
- **Maj_vote:** Using the major voting as $f$ in MCTS (e.g. MCTS-SC in §3.3);
- **Q_value:** Using the sum of Q-value in each path as $f$ in MCTS;
- **N_greedy:** At each step, select the node with the most frequent visits N and perform a top-down greedy search on the tree to obtain the final selected path;
- **Q_greedy:** At each step, select the node with the highest Q-value and perform a top-down greedy search on the tree to obtain the final selected path;
- **Oracle:** The coverage of the MCTS method.

In addition, we also use the consistency of the final answer output by the policy model itself as the source of the reward, denoted as 'Self'. The results are demonstrated in Figure 10. We can conclude that: (1) Even with the MCTS framework, the improvement in model reasoning brought by the RM is still minimal, further validating our conclusions in §2. (2) In Skywork and Skyworko1, the average performance of Reward is the best among all scoring functions. Therefore, in the MCTS-related experiments presented in the main text, we default to using it as the scoring function $f$.

Secondly, we focus on math reasoning in the main text, here we repeat our experiments on other types of reasoning tasks. Specifically, for math reasoning, we select another dataset: AQuA [13]. For commonsense reasoning, we select WinoGrande (WINO) [26] and CSQA [33]; For logical reasoning, we select ProofWriter [32] and ProntoQA [28] The results are demonstrated in Figure 11, 12, 13, 14 and 15. Lastly, we only use discriminative RM in the main text. All of these results are consistent with the conclusion in the main text.

# C Additional Experiments on Question Difficulty Approximation

In the main text, we calculate the question difficulty with assuming oracle access to a ground truth. However, in real-world applications, we are only given access to test prompts and do not know the true answers. Thus, we need to find a function that effectively estimates the problem difficulty without requiring ground truth. Specifically, we propose the following functions:

- **Length:** The average length of all responses to the question;
- **Count:** The count of different answers to the question;
- **Null:** The number of responses that fail to correctly generate the answer.

We classify the problems according to the difficulty levels as outlined in the main text and calculate the above three metrics across different levels of problem difficulty to compare the degree of correlation. The results are illustrated in Figure 16, 17 and 18. We can observe that, comparatively, the Count function is most directly proportional to difficulty. Therefore, we use this function to estimate difficulty when designing the CRISP method in §4.1.

## D  Additional Experiments across Different Difficulty Levels

In the main text, we only analyze the impact of question difficulty on the MATH dataset. To demonstrate the generalizability of our conclusions, we repeat this experiment on GSM8K [5] and Olympiadbench [9]. The former dataset contains 8.5K linguistically diverse elementary school math problems designed to evaluate arithmetic reasoning consistency, while the latter is an Olympiad-level bilingual multimodal scientific benchmark. Compared to MATH, the former is simpler, while the latter is more challenging. The results are illustrated in Table 5, 6 and 7. We can observe that the issues identified in Cl.1 are prevalent across various reasoning datasets.

## E  Case Analysis of Sampling Numbers Experiment

We start with a case analysis to uncover the issues inherent in the reward model. In the analysis, we randomly select five questions from different methods and examine the correctness of answers as $n$ scales. If a question is answered correctly, it indicates that the RM can accurately distinguish the positive examples from the negative ones, otherwise, it cannot. The results of this experiment are demonstrated in Figure 19, from which we can deduce that: **As $n$ increases, LLMs can generate incorrect responses that become increasingly challenging for the reward model to differentiate.** For some cases (like index 3 and 4 in Figure 19), RM assigns the highest score to newly generated incorrect responses, transforming the originally correct answers into incorrect ones.

## F  Cause Analysis of Temperature-Induced Accuracy Drop

We further conduct statistical analyses to uncover the reasons for this issue. For each $T$, we calculate the information entropy of incorrect answers across 16 samplings and report the distribution over 200 questions in Figure 21. As the temperature rises, the entropy for both models shows a gradually increasing trend, hence, the distribution of these negative samples becomes more random. This indicates that the policy model generates a greater number of low-frequency incorrect answers at higher temperatures. According to Cl.2, RM struggles to differentiate these negative examples from correct ones, leading to lower inference accuracy. This result not only elucidates the reasons behind the subpar performance of BoN under high diversity conditions but also further corroborates the inverse long-tail phenomenon of the RM.

## G  Diversity Experiment on Exploration Constant

In MCTS, apart from the tree structure, the explore weight $c$ also plays a crucial role in balancing the trade-off between exploitation (i.e. choosing actions that are known to yield high rewards) and exploration. A higher value of $c$ encourages more exploration, increasing the weight of the uncertain actions in the UCB formula. A lower value of $c$ favors exploitation, as it prioritizes actions with known higher rewards. We compare the MCTS performance under different $c$ and present the result in Figure 22. We can observe that an excessively large $c$ reduces performance (e.g. $c = 10.0$), indicating that overly high sampling diversity impairs reasoning accuracy, which is consistent with Cl.3 in our main text.

## H  Implementation Details in the Main Experiments

Here we provide a detailed account of the implementation specifics from the main experiments:

For Self-Consistency, we generate 32 samples and choose the major voting answer as the final prediction. For BoN, we set the temperature to 0.7 to control the diversity and choose the best answer from 32 samples. For BoN Weighted, we normalize the RM's scoring and use this score as a weight to conduct a weighted vote among different answers, selecting the final prediction. For MCTS, we set the rollout number to 16, the width to 5, the max depth to 5, and the explore weight to 0.1. For Beam Search, we set the Beam numbers to 8, the beam width to 5, and the max depth to 5.

For our method, we generate 16 samples with a temperature setting of 0.7 in the first iteration. In subsequent iterations, we set the sampling numbers to 8 for ORM, 4 for PRM, and the max depth to

3. In prefix extraction, for ORM, we select the top-1 path, for PRM, we select the top-2 paths. For the evaluation data, we sample 500 questions from GSM8K and MATH-500, while sampling 200 questions from OlympiadBench.

We release the prompts we use in Table 8, 9, 10, 11, 12 and 13. All experiments were conducted on NVIDIA A100 GPUs.

# I   Ablation Study

To verify the effectiveness of each module of CRSIP, we conduct ablation experiments using 200 samples from GSM8K and MATH generated by Qwen2.5-3B. The experimental settings are as follows:

- **w/o Termination:** Disable the early termination condition based on the number of clusters;
- **w/o Aggregation:** Eliminate the clustering operation and use the score of each path instead of cluster scores for selection (similar to MCTS);
- **w/o Prefixing:** Cancel the operation of directly generating the remaining steps according to the prefix set, and instead generate intermediate nodes layer by layer (similar to MCTS and Beam).

Figure 23 shows the result of the ablation study. Removing each component leads to a decline in performance. Specifically, although w/o termination causes only a small drop, its inclusion not only improves performance but also reduces inference time.

# J   Cost Analysis

We use Qwen2.5-3B as the policy model and Shepherd-PRM [35] as the reward model, and compare the inference time and token usage of different algorithms across various tasks. Each algorithm is required to perform 5 rollouts on the same devices, and the average is computed across all test instances. The results in Figure 24 demonstrate that our method is highly efficient. It achieves up to a **66%** reduction in inference time compared to advanced RM-integrated methods like MCTS and Beam Search, while preserving the runtime and token efficiency of the basic BoN method.

| Reward Model | Score | Chat | Chat Hard | Safety | Reasoning |
|---|---|---|---|---|---|
| Skywork-Reward-Llama-3.1-8B | 93.1 | 94.7 | 88.4 | 92.7 | 96.7 |
| ArmoRM-Llama3-8B-v0.1 | 89.0 | 96.9 | 76.8 | 92.2 | 97.3 |
| Gemini-1.5-pro-0514 | 88.1 | 92.3 | 80.6 | 87.5 | 92.0 |
| gpt-4-0125-preview | 84.3 | 95.3 | 74.3 | 87.2 | 86.9 |
| Meta-Llama-3-70B-Instruct | 75.4 | 97.6 | 58.9 | 69.2 | 78.5 |

Table 3: Comparison of RM's performance on RewardBench.

| Model | GSM8K | MATH | OlympiadBench | OmniMATH | Average |
|---|---|---|---|---|---|
| Shepherd-PRM-7B | 47.9 | 29.5 | 24.8 | 23.8 | 31.5 |
| Skyworko1-PRM-7B | 70.8 | 53.6 | 22.9 | 21.0 | 42.1 |
| Meta-Llama-3-70B-Instruct | 52.2 | 22.8 | 21.2 | 20.0 | 29.1 |
| Llama-3.1-70B-Instruct | 74.9 | 48.2 | 46.7 | 41.0 | 52.7 |
| Qwen2-72B-Instruct | 67.6 | 49.2 | 42.1 | 40.2 | 49.8 |

Table 4: Comparison of RM's performance on ProcessBench.

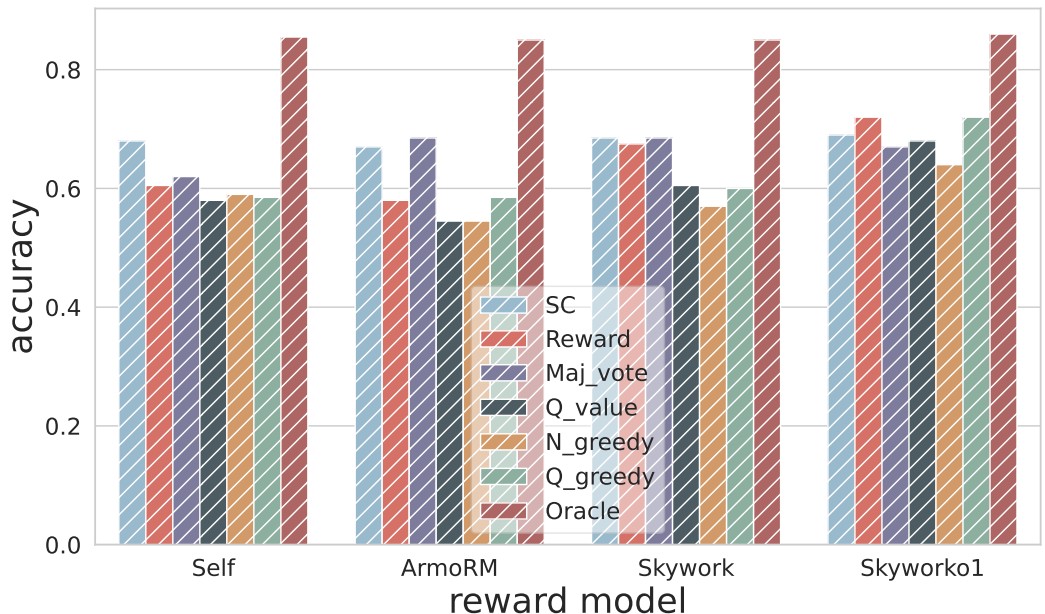

Figure 10: The performance of different reward models using the MCTS inference on the MATH dataset ($n = 16$, Qwen-2.5-3B).

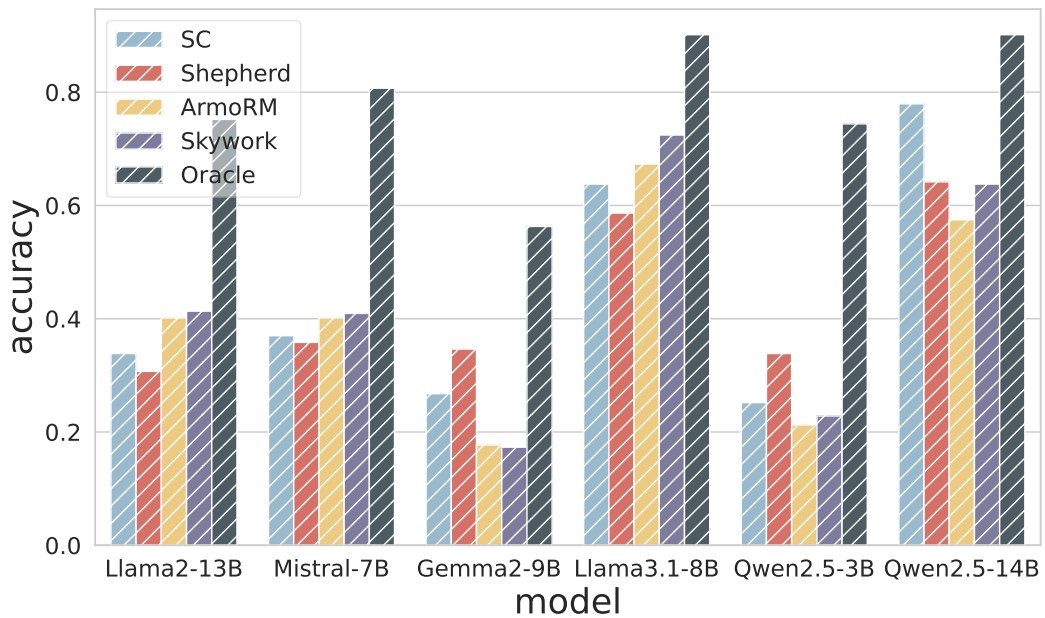

Figure 11: The performance of different policy models using various reward models for BoN inference on the AQuA dataset ($n = 10$).

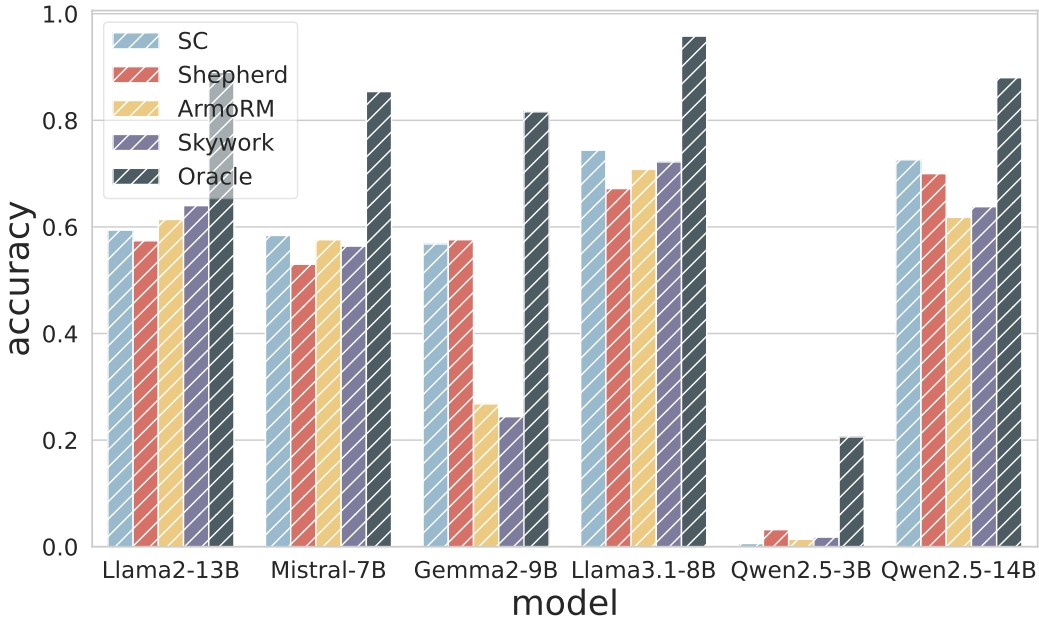

Figure 12: The performance of different policy models using various reward models for BoN inference on the WinoGrande dataset ($n = 10$).

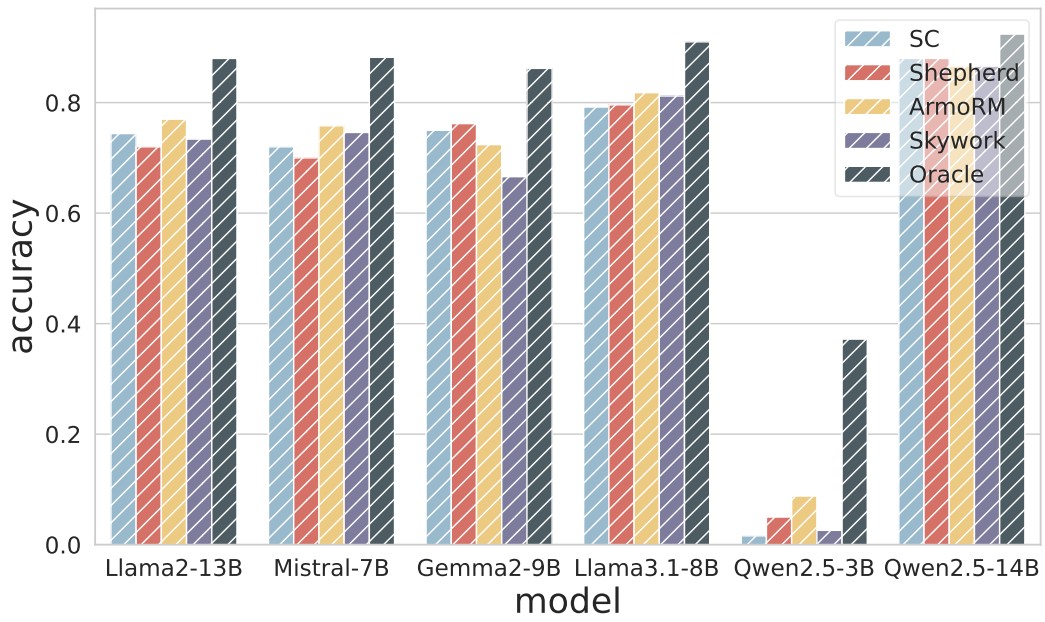

Figure 13: The performance of different policy models using various reward models for BoN inference on the CSQA dataset ($n = 10$).

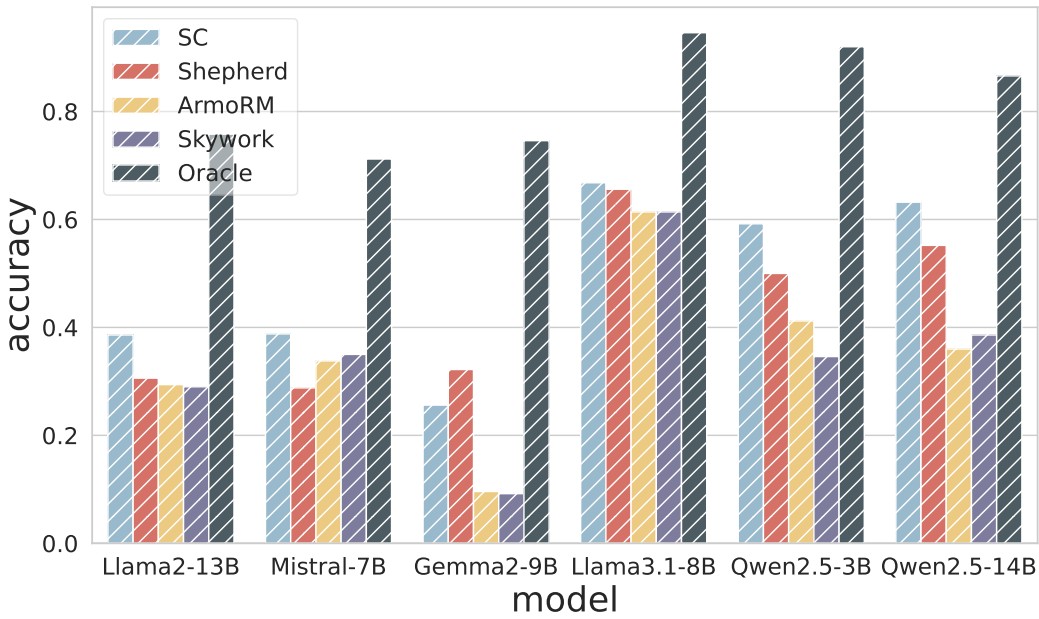

Figure 14: The performance of different policy models using various reward models for BoN inference on the ProofWriter dataset ($n = 10$).

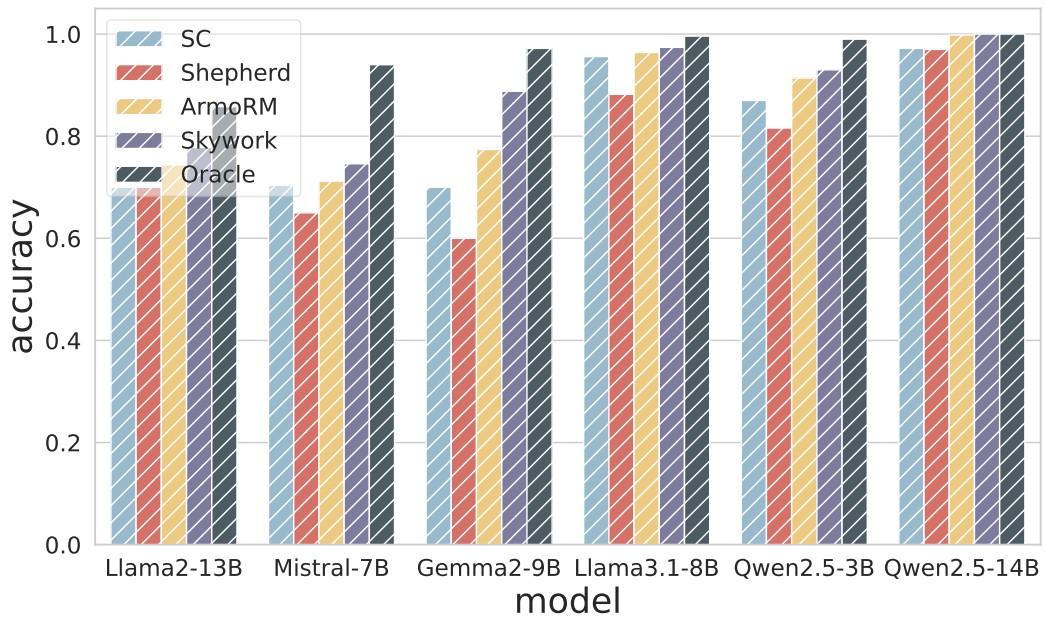

Figure 15: The performance of different policy models using various reward models for BoN inference on the ProntoQA dataset ($n = 10$).

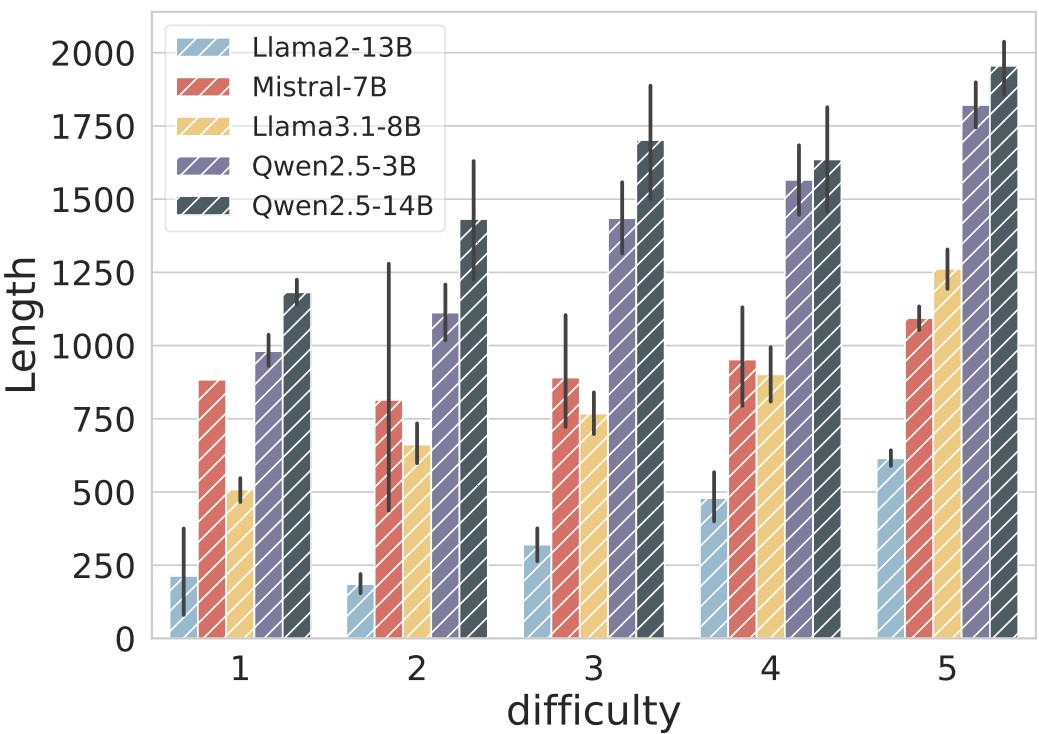

Figure 16: The correlation between output length and the question difficulty.

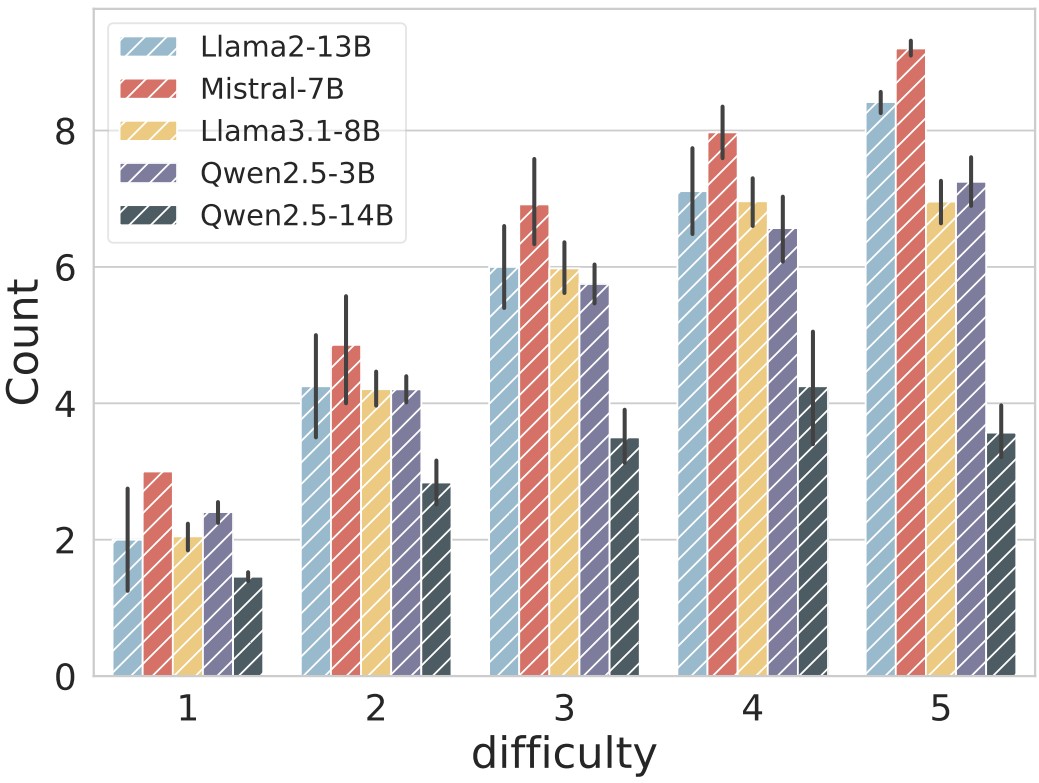

Figure 17: The correlation between the count of answers and the question difficulty.

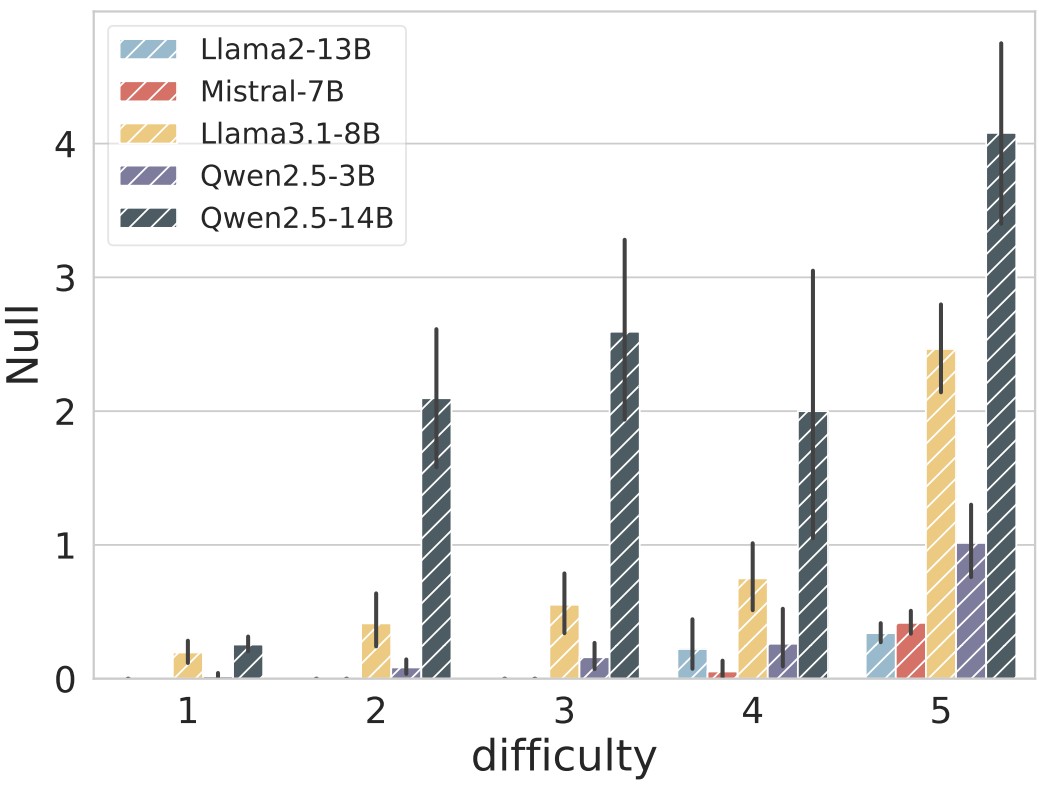

Figure 18: The correlation between the count of no answers and the question difficulty.

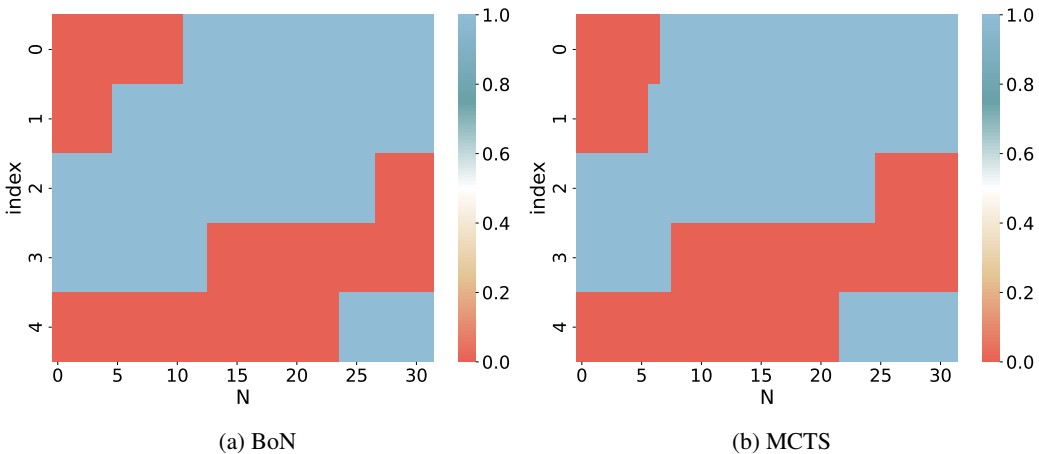

(a) BoN                    (b) MCTS

Figure 19: The variation in question answering correctness as the sampling number changes. Blue indicates a correct answer, while red indicates an incorrect answer.

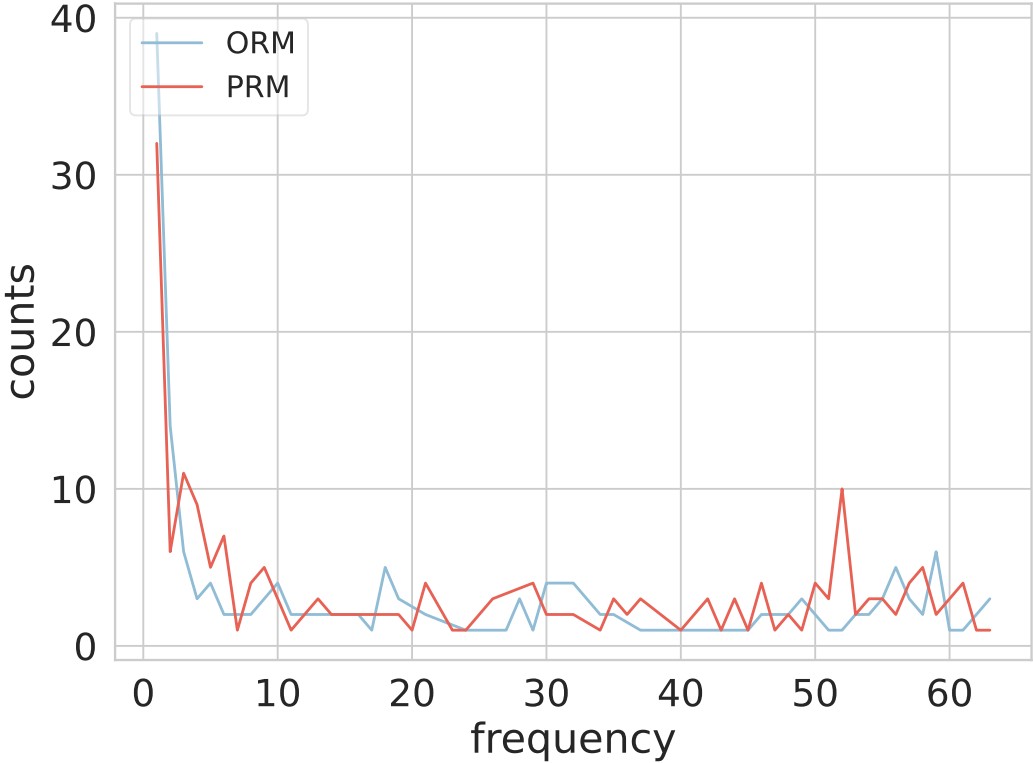

Figure 20: Frequency statistics of the highest-scored negative responses in MCTS.

Table 5: Comparison of performance across different difficulty levels on 500 samples of GSM8K (Qwen2.5-3B).

| Method | Level 1 | Level 2 | Level 3 | Level 4 | Level 5 | All |
|---|---|---|---|---|---|---|
| Self-Consistency(@128) | **99.7** | 96.8 | 80.0 | 34.6 | 3.2 | 83.2 |
| Best-of-128 + ORM | 98.0 | 87.1 | 72.0 | **65.4** | 12.9 | 83.8 |
| - SC | -1.7 | -9.7 | -8.0 | 30.8 | 9.7 | 0.6 |
| Best-of-128 + PRM | 98.3 | **100.0** | **96.0** | 57.7 | **30.6** | **87.8** |
| - SC | -1.4 | 3.2 | 16.0 | 23.1 | 27.4 | 4.6 |
| Count | 356 | 31 | 25 | 26 | 62 | 500 |

Table 6: Comparison of performance across different difficulty levels on MATH-500 (Qwen2.5-3B).

| Method | Level 1 | Level 2 | Level 3 | Level 4 | Level 5 | All |
|---|---|---|---|---|---|---|
| Self-Consistency(@128) | 98.8 | **98.8** | **80.4** | 49.2 | 5.3 | 65.4 |
| Best-of-128 + ORM | **99.4** | 92.8 | 69.6 | **58.5** | 17.3 | **67.8** |
| - SC | 0.6 | -6.0 | -9.8 | 9.3 | 12.0 | 2.4 |
| Best-of-128 + PRM | 88.3 | 71.1 | 78.6 | 53.8 | **21.8** | 62.2 |
| - SC | -10.5 | -27.7 | -1.8 | 4.6 | 16.5 | -3.2 |
| Count | 163 | 83 | 56 | 65 | 133 | 500 |

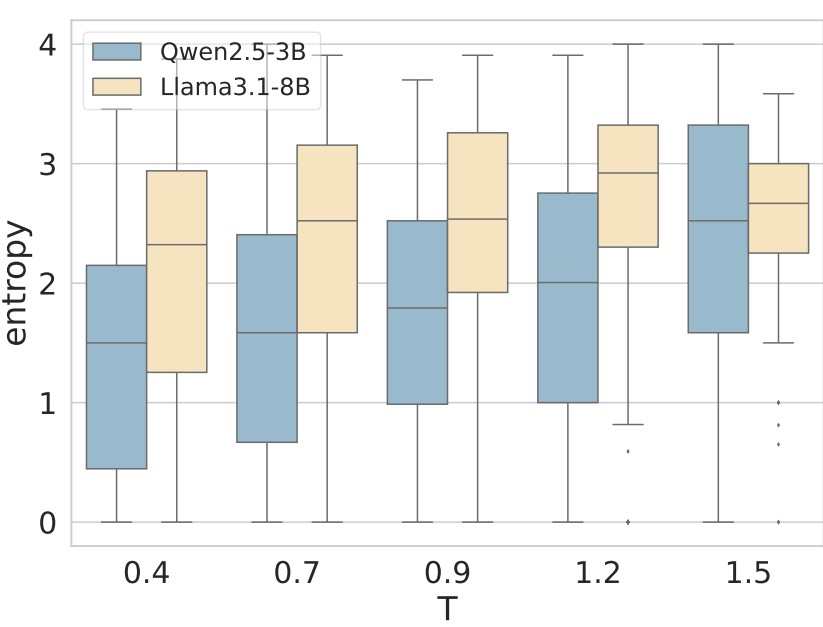

Figure 21: Information entropy of incorrect answers under different sampling temperatures.

Table 7: Comparison of performance across different difficulty levels on 200 samples of Olympiad-Bench (Qwen2.5-3B).

| Method | Level 1 | Level 2 | Level 3 | Level 4 | Level 5 | All |
|---|---|---|---|---|---|---|
| Self-Consistency(@32) | **100.0** | **100.0** | 64.3 | 50.0 | 0.8 | 30.5 |
| Best-of-32 + ORM | 100.0 | 80.0 | **78.6** | 40.0 | 3.8 | 31.5 |
| - SC | 0.0 | -20.0 | 14.3 | -10.0 | 3.0 | 1.0 |
| Best-of-32 + PRM | 100.0 | 100.0 | **78.6** | **50.0** | **6.9** | **34.0** |
| - SC | 0.0 | 0.0 | 14.3 | 0.0 | 6.1 | 3.5 |
| Count | 31 | 15 | 14 | 10 | 130 | 200 |

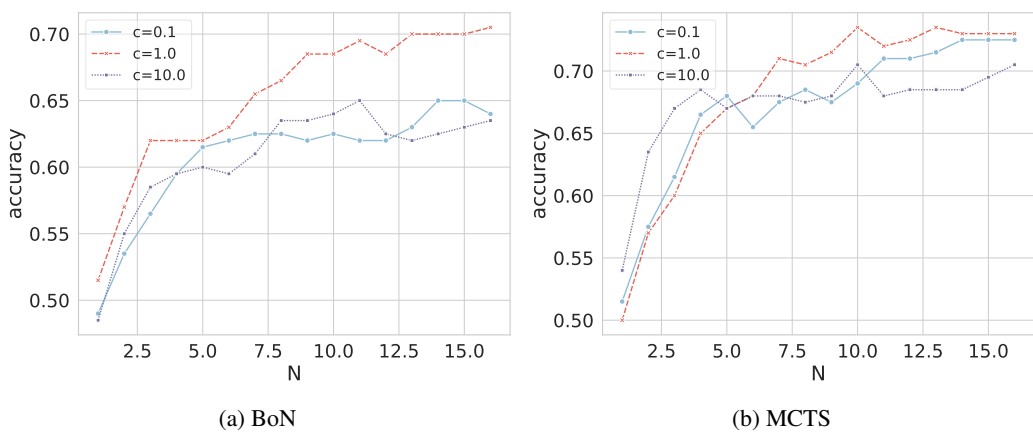

(a) BoN                    (b) MCTS

Figure 22: Performance comparison across different explore weight $c$ on Qwen2.5-3B.

**Algorithm 1** Clustered Reward Integration with Stepwise Prefixing

---

**Require:** Policy model $\mathcal{M}$, reward score $f$, question $q$, max steps $m$, sampling numbers $n$, top-$k$ parameter $k$

1:   $i \leftarrow 0$
2:   $\mathcal{R} \leftarrow \emptyset$                                           $\triangleright$ All responses
3:   $\mathcal{P} \leftarrow \emptyset$                                       $\triangleright$ Response prefixes
4:   $\mathcal{F} \leftarrow \emptyset$                                             $\triangleright$ Score map
5:   $\mathcal{C} \leftarrow \emptyset$                                               $\triangleright$ Clusters
6:   **while** $i < n$ **do**
7:      **if** $i = 0$ **then**
8:          $\mathcal{R} \leftarrow \mathcal{M}(q, n)$                           $\triangleright$ Generate $n$ initial responses
9:          **if** $|\operatorname{Cluster}(\mathcal{R})| = 1$ **then**
10:             **return** $\mathcal{R}[0]$                   $\triangleright$ Early exit if only one cluster
11:          **end if**
12:      **else**
13:          $\mathcal{R}_{\text{top}} \leftarrow \left\{ \arg\max_{r \in \mathcal{C}_j} f(r) \,\middle|\, \mathcal{C}_j \in \mathcal{C}_{\text{top}} \right\}$
14:          $\mathcal{P} \leftarrow \{ r[:i+1] \mid r \in \mathcal{R}_{\text{top}} \}$            $\triangleright$ Truncate top responses
15:          $\mathcal{R} \leftarrow \mathcal{R} \cup \mathcal{M}(q, n, \mathcal{P})$         $\triangleright$ Decode more based on prefixes
16:      **end if**
17:      $\mathcal{C} \leftarrow \operatorname{Cluster}(\mathcal{R})$                      $\triangleright$ Cluster current responses
18:      **for all** $\mathcal{C}_j \in \mathcal{C}$ **do**
19:          $\mathcal{F}(\mathcal{C}_j) \leftarrow \sum_{x \in \mathcal{C}_j} f(x)$            $\triangleright$ Assign cluster-wise reward
20:      **end for**
21:      $\mathcal{C}_{\text{top}} \leftarrow$ top-$k$ responses in $\mathcal{C}$ by $\mathcal{F}$
22:      $i \leftarrow i + 1$
23:   **end while**
24:   **return** $\mathcal{R}_{\text{top}}[0]$

---

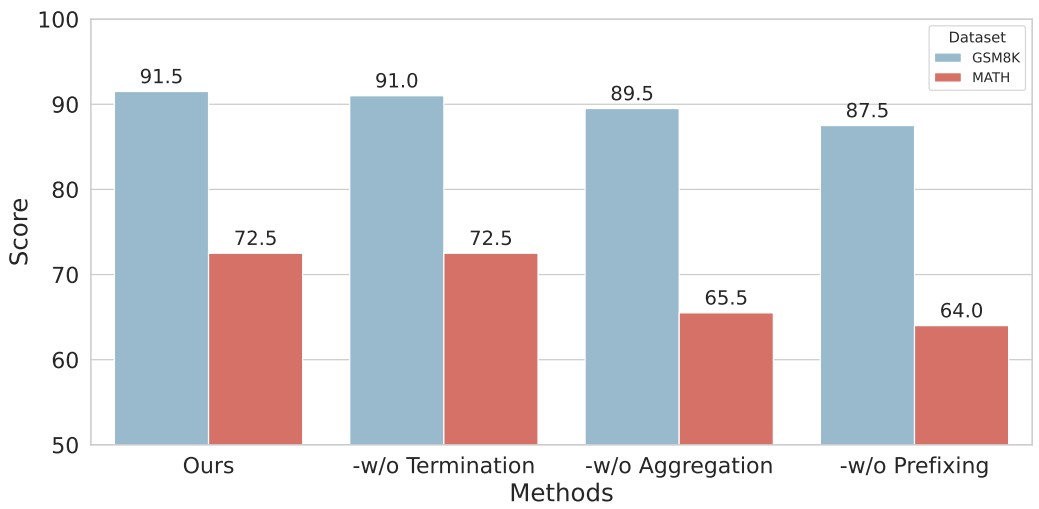

Figure 23: Results of our ablation study.

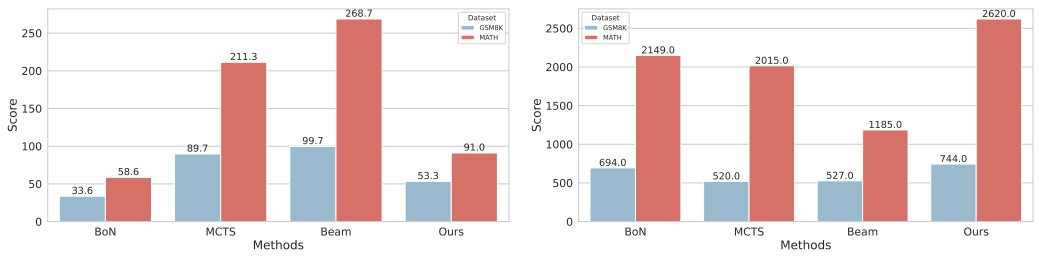

(a) Time Consumption Comparison (s)      (b) Token Consumption Comparison

Figure 24: Results of our cost analysis.

Table 8: Prompts used to sample reasoning paths on the GSM8K dataset.

**Prompt**

Please act as a math teacher and solve the math problem step by step. At the final step, a conclusive answer is given in the format of "The answer is: \boxed{<ANSWER>}.", where <ANSWER> should be a numeric answer.

**# Question:**
Mr. Ruther sold $\frac{3}{5}$ of his land and had 12.8 hectares left. How much land did he have at first?
**# Reasoning:**
Step 1: Mr. Ruther is left with $1 - \frac{3}{5} = \frac{2}{5}$ of his land.
Step 2: Since $\frac{2}{5}$ equals 12.8 hectares, then $\frac{1}{5} = \frac{12.8}{2} = 6.4$ hectares.
Step 3: Total land $= 6.4 \times 5 = 32$ hectares.
Step 4: The answer is: $\boxed{32}$

**# Question:**
The Doubtfire sisters are driving home with 7 kittens adopted from the local animal shelter when their mother calls to inform them that their two house cats have just had kittens. She says that Patchy, the first cat, has had thrice the number of adopted kittens, while Trixie, the other cat, has had 12. How many kittens does the Doubtfire family now have?
**# Reasoning:**
Step 1: Patchy has had $3 \times 7 = 21$ kittens.
Step 2: Trixie has had 12 kittens. Total from both cats $= 21 + 12 = 33$.
Step 3: Total kittens including adopted $= 7 + 33 = 40$.
Step 4: The answer is: $\boxed{40}$

**# Question:**
After transferring to a new school, Amy made 20 more friends than Lily. If Lily made 50 friends, how many friends do Lily and Amy have together?
**# Reasoning:**
Step 1: Amy made $50 + 20 = 70$ friends.
Step 2: Total friends $= 70 + 50 = 120$.
Step 3: The answer is: $\boxed{120}$
**# Question:**
{current question}
**# Reasoning:**

Table 9: Prompts used to sample reasoning paths on the MATH dataset.

**Prompt**

Please act as a math teacher and give step-by-step solutions to the user's questions. At the final step, a conclusive answer is given in the format of "The answer is: <ANSWER>.", where <ANSWER> should be a numeric answer.

**# Question:**

How many 3-letter words can we make from the letters A, B, C, and D, if we are allowed to repeat letters, and we must use the letter A at least once? (Here, a word is an arbitrary sequence of letters.)

**# Reasoning:**

Step 1: There are $4^3$ three-letter words from A, B, C, and D, and there are $3^3$ three-letter words from just B, C, and D.

Step 2: There must, then, be $4^3 - 3^3 = 64 - 27 = \boxed{37}$ words from A, B, C, and D containing at least one A.

Step 3: The answer is: $\boxed{37}$

**# Question:**

In the diagram, square $ABCD$ has sides of length 4, and $\triangle ABE$ is equilateral. Line segments $BE$ and $AC$ intersect at $P$. Point $Q$ is on $BC$ so that $PQ$ is perpendicular to $BC$ and $PQ = x$.

**# Reasoning:**

Step 1: Since $\triangle ABE$ is equilateral, we know that $\angle ABE = 60°$.

Step 2: Therefore,
$$\angle PBC = \angle ABC - \angle ABE$$
$$= 90° - 60° = 30°.$$

Step 3: Since $AB = BC$, we know that $\triangle ABC$ is a right isosceles triangle and $\angle BAC = \angle BCA = 45°$.

Step 4: Then, $\angle BCP = \angle BCA = 45°$ and
$$\angle BPC = 180° - \angle PBC - \angle BCP$$
$$= 180° - 30° - 45° = \boxed{105°}.$$

Step 5: The answer is: $\boxed{105}$

**# Question:**

Find the *positive* real number(s) $x$ such that

$$\frac{1}{2}(3x^2 - 1) = (x^2 - 50x - 10)(x^2 + 25x + 5).$$

**# Reasoning:**

Step 1: Write $a = x^2 - 50x - 10$ and $b = x^2 + 25x + 5$.

Step 2: Then the equation given becomes

$$\frac{a + 2b - 1}{2} = ab,$$

so $0 = 2ab - a - 2b + 1 = (a - 1)(2b - 1)$.

Step 3: Then $a - 1 = x^2 - 50x - 11 = 0$ or $2b - 1 = 2x^2 + 50x + 9 = 0$.

Step 4: The former has a positive root, $x = \boxed{25 + 2\sqrt{159}}$, while the latter does not.

Step 5: The answer is: $\boxed{25 + 2\sqrt{159}}$

**# Question:**

{current question}

**# Reasoning:**

Table 10: Prompts used to sample reasoning paths on the Olympiadbench dataset.

**Prompt**

Please act as a math teacher and give step-by-step solutions to the user's questions. At the final step, a conclusive answer is given in the format of "The answer is: \boxed{<ANSWER>}.", where <ANSWER> should be a numeric answer.

**# Question:**
Let $T$ be a rational number. Compute $\sin^2 \frac{T\pi}{2} + \sin^2 \frac{(5-T)\pi}{2}$.
**# Reasoning:**
Step 1: Note that $\sin \frac{(5-T)\pi}{2} = \cos\left(\frac{\pi}{2} - \frac{(5-T)\pi}{2}\right) = \cos\left(\frac{T\pi}{2} - 2\pi\right) = \cos\frac{T\pi}{2}$.
Step 2: Thus the desired quantity is $\sin^2 \frac{T\pi}{2} + \cos^2 \frac{T\pi}{2} = \boxed{1}$.
Step 3: The answer is: $\boxed{1}$

**# Question:**
Let $T = 11$. Compute the value of $x$ that satisfies $\sqrt{20 + \sqrt{T + x}} = 5$.
**# Reasoning:**
Step 1: Squaring both sides gives $20 + \sqrt{T + x} = 25$, so $\sqrt{T + x} = 5$.
Step 2: Squaring again gives $T + x = 25$, so $x = 25 - T = 14$.
Step 3: The answer is: $\boxed{14}$

**# Question:**
The sum of the interior angles of an $n$-gon equals the sum of the interior angles of a pentagon plus the sum of the interior angles of an octagon. Compute $n$.
**# Reasoning:**
Step 1: The sum of interior angles of an $n$-gon is $180°(n-2)$.
Step 2: A pentagon has sum $180°(5-2) = 540°$, and an octagon has sum $180°(8-2) = 1080°$.
Step 3: So $180(n-2) = 540 + 1080 = 1620$, hence $n - 2 = 9$, so $n = 11$.
Step 4: The answer is: $\boxed{11}$
**# Question:**
{current question}
**# Reasoning:**

Table 11: Prompts used to sample reasoning paths on the CSQA dataset.

**Prompt**

Please act as a commonsense teacher and solve the commonsense reasoning problem step by step.

**# Question:**
Google Maps and other highway and street GPS services have replaced what?
**# Options:**
(A) atlas    (B) mexico    (C) countryside    (D) united states    (E) oceans
**# Reasoning:**
Step 1: Electronic maps and GPS services are the modern version of paper atlas.
Step 2: In that case, the atlas have been replaced by Google Maps and other highway and street GPS services.
Step 3: The answer is: **A**

**# Question:**
You can share files with someone if you have a connection to a what?
**# Options:**
(A) freeway    (B) radio    (C) wires    (D) computer network    (E) electrical circuit
**# Reasoning:**
Step 1: Files usually can be stored in the computers.
Step 2: In that case, we can share them over the Internet.
Step 3: Thus, if we connect to a computer network, we can share the file with others.
Step 4: The answer is: **D**

**# Question:**
The fox walked from the city into the forest, what was it looking for?
**# Options:**
(A) pretty flowers    (B) hen house    (C) natural habitat    (D) storybook    (E) dense forest
**# Reasoning:**
Step 1: Since the fox walk from the city into the forest, he may looks for something in the forest but not in the city.
Step 2: From all of the options, the natural habitat are usually away from cities.
Step 3: The answer is: **C**
**# Question:**
{current question}
**# Options:**
{current options}
**# Reasoning:**

Table 12: Prompts used to sample reasoning paths on the SIQA dataset.

**Prompt**

Please act as a commonsense teacher and solve the commonsense reasoning problem step by step.

**# Question:**
Quinn wanted to help me clean my room up because it was so messy. What will Quinn want to do next?
**# Options:**
(A) Eat messy snacks     (B) help out a friend     (C) Pick up the dirty clothes
**# Reasoning:**
Step 1: Quinn wants to clean the room up.
Step 2: Picking up the dirty clothes is one way to clean the room.
Step 3: Thus, Quinn will want to pick up the dirty clothes next.
Step 4: The answer is: C

**# Question:**
Sydney had so much pent up emotion, they burst into tears at work. How would Sydney feel afterwards?
**# Options:**
(A) affected     (B) like they released their tension     (C) worse
**# Reasoning:**
Step 1: Crying is often a way to release tension.
Step 2: Sydney burst into tears at work.
Step 3: Thus, she would release the tension.
Step 4: The answer is: B

**# Question:**
Their cat kept trying to escape out of the window, so Jan placed an obstacle in the way. How would Jan feel afterwards?
**# Options:**
(A) scared of losing the cat     (B) normal     (C) relieved for fixing the problem
**# Reasoning:**
Step 1: The cat tried to escape so Jan needed to stop it to avoid losing the cat.
Step 2: Jan placed an obstacle in the way so the cat could not escape.
Step 3: The problem has been solved.
Step 4: Thus, Jan will feel relieved for fixing the problem.
Step 5: The answer is: C
**# Question:**
{current question}
**# Options:**
{current options}
**# Reasoning:**

Table 13: Prompts used to sample reasoning paths on the LogiQA dataset.

**Prompt**

Please act as a logical teacher and reason step by step to solve the logical reasoning problem.

**# Context:**
There are 90 patients with a disease T that is very difficult to treat and has taken the same routine drug. The patients were divided into two equal groups. The first group was given an experimental drug W, which is used to treat T, and the second group was given a placebo without W. Statistics ten years later showed that 44 people died in both groups, so the experimental drug was ineffective.

**# Question:**
Based on the above information, which of the following options, if correct, will best weaken the above argument?

**# Options:**
(A) Among the patients who died above, the average year of death in the second group was two years earlier than that in the first group.    (B) Among the patients who died, the average life span of the second group was two years younger than that of the first group.    (C) Among the above-mentioned living patients, the condition of the second group was more serious than that of the first group.    (D) Among the above-mentioned living patients, those in the second group were older than those in the first group.

**# Reasoning:**
Step 1: Analyzing each option: A suggests drug W might extend life since the average death year in the drug W group is later than the placebo, directly challenging the drug's perceived ineffectiveness.
Step 2: B, similar to A, implies longer life in the drug W group but doesn't directly link to post-treatment lifespan.
Step 3: C indicates drug W may reduce disease severity but doesn't address lifespan or mortality, the main focus.
Step 4: D, about age differences, lacks direct relevance to drug effectiveness.
Step 5: Therefore, A most effectively weakens the argument against drug W's effectiveness.
Step 6: The answer is: A

**# Question:**
{current question}
**# Options:**
{current options}
**# Reasoning:**

