# OpenReview forum: "Fixing the Broken Compass: Diagnosing and Improving Inference-Time Reward Modeling"
_NeurIPS.cc/2025/Conference — Submitted to NeurIPS 2025_

### Official Review · Reviewer_HDQp · 2025-07-01

**Clarity:** 3
**Significance:** 2
**Originality:** 2
**Rating:** 4
**Confidence:** 3

**Summary:**

The paper provides analysis on limitations of using inference-time reward models (RM), specifically on how it underperforms simple majority vote aggregation from sampling under various settings, and proposed a method that first clusters reasoning paths based on final answer, aggregates reward signal at the cluster level, before updating prefix prompts adaptively to guide generation.

**Questions:**

Please see weaknesses for questions. Would appreciate clarifications and elaboration.

**Ethical Concerns:**

["NO or VERY MINOR ethics concerns only"]

**Final Justification:**

The authors have partially addressed my concerns, hence I maintain my score.

**Limitations:**

Please see weaknesses.

**Paper Formatting Concerns:**

nil

**Quality:**

3

**Strengths And Weaknesses:**

**Strengths**
- The paper analyzes an important problem of inference-time reward modeling, and took a closer look at when such methods actually perform worse than simpler methods such as majority voting aggregation that does not require the use of reward models.
- The empirical results presented in Table 1 seem to indicate that the proposed method does perform better than the baselines considered in the paper.

**Weaknesses**
- In many of the results tables and charts, the performance differences are fairly close -- it would be useful for there to be error bars indicated to provide a better sense of whether performance gains are signfiicant.
- Some of the findings could be better explained or compared across each other. For example, Fig.7 seems to show that the RM based approaches outperform the SC baseline across all temperatures, which appears at odds with the trend indicated in Fig 1. This also relates to the question of whether after including error bars in the results, the implications remain the same.
- It would be useful to further expand discussion on related works or include in baselines additional inference-time optimization methods, such as those that provide further improvements even without the use of reward models while adopting majority vote approaches. For example, works on optimizing LLM ensembles based on the same base model to improve reasoning tasks such as [1-3] would provide useful comparisons.
- Further elaboration of the cost analysis should be included in the main paper, as it is an important consideration factor. Fig. 24 in the appendix does not seem to support the claim that the proposed method incur lower costs -- for example, it shows that the proposed method has significantly higher token consumption, and more time than BoN.

[1] Lau et al, "Dipper: Diversity in Prompts for Producing Large Language Model Ensembles in Reasoning Tasks", NeurIPS Workshop on Foundation Model Interventions 2024

[2] Chen et al, "AgentVerse: Facilitating Multi-Agent Collaboration and Exploring Emergent Behaviors.", ICLR 2024

[3] Du et al, "Improving factuality and Reasoning in Language Models through Multiagent Debate, ICML 2024

---

> ### Author Rebuttal · Authors · 2025-07-30
>
> + **Weakness 1:**  It would be useful for there to be error bars indicated to provide a better sense of whether performance gains are significant.
> + **Reply 1:**  We greatly appreciate your suggestions, as they are very helpful in improving our paper. Since our work involves many complex inference algorithms, the inference process often requires considerable time and computational resources. Therefore, most of the experiments in the main text were run only once.  To demonstrate that our findings and method improvements are significant, we conducted experiments across different parameters, models, and datasets. As you can see from the charts and tables, our conclusions and methods are effective across different settings. We commit to adding error bars in the subsequent version of our paper to further enhance its reliability.
>
>
>
> + **Weakness 2:** Some of the findings could be better explained or compared across each other. Whether after including error bars in the results, the implications remain the same.
>
> + **Reply 2:** We apologize for the misunderstanding caused by the unclear presentation of some experimental details in our work. In fact, **the two experimental setups in Figure 1 and Figure 7 are completely different**, so they are not suitable for direct comparison. For the former, we set the temperature to 1.0 and sampled each question 10 times (as indicated in **the caption of Figure 1**), testing on the entire MATH dataset. For the latter, we sampled each question 16 times and tested on 200 samples from the MATH dataset (see **lines 562-564** in Appendix F). It is reasonable for different parameter settings to lead to variations in their respective performances.
>
>   On the other hand, this seemingly contradictory conclusion is unrelated to the focus of our work and does not affect any of the conclusions of the paper. As stated in **lines 92-93**, the experiment in Figure 1 is intended to demonstrate that **the performance gap between the existing RM's BoN and SC is small, indicating room for improvement, rather than to evaluate its absolute performance.** Therefore, the lack of a significant difference between BoN and SC further highlights the shortcomings of the RM and underscores the need for optimization.
>
>
>
> + **Weakness 3:** It would be useful to further expand discussion on related works or include in baselines additional inference-time optimization methods.
>
> + **Reply 3:** Thank you for your suggestions. In this work, **we primarily focus on reward-based inference optimization in the test-time scaling scenario** (see lines 17–23). Therefore, traditional prompt-based methods such as multi-agent discussions are not within the scope of our study or comparisons. To further demonstrate the effectiveness of our method, here we reproduce the approaches **from [2,3]** on Qwen2.5-3B and compare the performance. The results on GSM8K and MATH are as follows:
>
>   | Method           | GSM8K     | MATH      |
>   | ---------------- | --------- | --------- |
>   | AgentVerse       | 0.642     | 0.528     |
>   | MultiagentDebate | 0.885     | 0.644     |
>   | **Ours**         | **0.908** | **0.698** |
>
>   It demonstrates that our method significantly outperforms the multi-agent discussion paradigm. We hypothesize that this gap is due to the insufficient reflective capability of the generative model. Although open-source models like Qwen2.5 perform reasonably well on mathematical tasks, their relatively smaller parameter size still limits their discriminative and reflective abilities [4,5]. As a result, they cannot achieve the same level of improvement through multiple rounds of discussion as larger, closed-source models like GPT-4.  Our RM-based inference method can be viewed as introducing an RM as a role in multi-agent discussions. Since these RMs are often trained extensively on discriminative tasks, **they can provide higher-quality reflective signals compared to the generative model itself**, helping to improve the final output. We promise to discuss these related works and give a more comprehensive analysis in the subsequent version of the paper.
>
>
> + **Weakness 4:** Further elaboration of the cost analysis should be included in the main paper. Fig. 24 in the appendix does not seem to support the claim that the proposed method incur lower costs.
>
> + **Reply 4:** We apologize for the lack of clarity in our cost analysis in the main paper. Here, **we primarily compare the cost of CRISP with two search algorithms, MCTS and Beam Search**, while treating **BoN as a lower bound** on cost, as it represents the simplest direct sampling method.
>
>   + In terms of time cost, we are significantly lower than the other two methods. According to Figure 24(a), in mathematics tasks, MCTS takes **2.3 times** longer than our method, while Beam Search takes nearly **3 times** as long. Moreover, our time cost is close to that of BoN, staying within approximately **1.x times** on both datasets.
>
>   + In terms of token cost, our method does consume more tokens compared to the other two methods. This is partly because, when counting token usage, we only consider the average number of tokens in the final complete reasoning paths. **For MCTS and Beam Search, many intermediate paths are generated during the search but later discarded, and these are not accounted for in the token statistics.**  In contrast, CRISP fully explores and generates all paths, which are included in the count. Taking this factor into account, and considering that **our token cost is not much higher than BoN (approximately 1.2×)**, we believe this level of cost is acceptable.
>
>   In summary, considering that the primary concern for downstream inference cost is time, our method demonstrates significantly lower cost compared to the other advanced search algorithms (MCTS and Beam Search). Additionally, since our approach only incurs a modest increase in token usage, and delivers the best performance, we believe **it achieves a strong balance between efficiency and effectiveness**.
>
>
>
> [1] Lau et al. Dipper: Diversity in Prompts for Producing Large Language Model Ensembles in Reasoning Tasks.
>
> [2] Chen et al. AgentVerse: Facilitating Multi-Agent Collaboration and Exploring Emergent Behaviors.
>
> [3] Du et al. Improving factuality and Reasoning in Language Models through Multiagent Debate.
>
> [4] Peter West et al. The Generative AI Paradox: “What It Can Create, It May Not Understand”.
>
> [5] Jie Huang et al. Large Language Models Cannot Self-Correct Reasoning Yet.

---

> > ### Comment · Reviewer_HDQp · 2025-08-04
> >
> > Thanks for the detailed response.
> >
> > The authors may consider running paired t-tests if the intention is to demonstrate that across models and datasets, their method can outperform each of the baselines -- given the close gap in the performance between the proposed method and the baselines, this may provide additional support for the claimed performance gains.

---

> > > ### Author Response · Authors · 2025-08-05
> > > **Reply to Reviewer HDQp**
> > >
> > > Thank you for your valuable suggestions, which have significantly improved our work. Here, we repeat the experiments on the MATH dataset (200 samples) using Qwen-2.5-3B combined with Skywork-o1 for three runs. The results are as follows:
> > >
> > > | Method      | 1    | 2    | 3    |
> > > | ----------- | ---- | ---- | ---- |
> > > | BoN         | 0.61 | 0.64 | 0.63 |
> > > | Weighted SC | 0.63 | 0.71 | 0.67 |
> > > | MCTS        | 0.71 | 0.72 | 0.73 |
> > > | Beam Search | 0.72 | 0.70 | 0.72 |
> > > | **Ours**    | 0.76 | 0.78 | 0.77 |
> > >
> > > Next, we conduct paired t-tests based on these results, and the outcomes are as follows:
> > >
> > > | Comparison          | t-statistic | p-value  |
> > > | ------------------- | ----------- | -------- |
> > > | BoN vs Ours         | -43.000000  | 0.000540 |
> > > | Weighted SC vs Ours | -5.773503   | 0.028714 |
> > > | MCTS vs Ours        | -8.660254   | 0.013072 |
> > > | Beam vs Ours        | -4.714952   | 0.042159 |
> > >
> > > As observed, all p-values are less than 0.05, suggesting that the performance improvement of our method over the baselines is statistically significant. We apologize that, due to the time constraints of the rebuttal period, we were unable to repeat the t-tests across all models and benchmarks. But we commit to include full experiments and details in the subsequent version of the paper. If you are interested in any additional experimental results, please feel free to let us know.
> > >
> > > We welcome further discussion to better understand any remaining issues. Thank you again for your time and suggestions.

---

> > > > ### Author Response · Authors · 2025-08-07
> > > > **Reply to Reviewer HDQp**
> > > >
> > > > Dear Reviewer HDQp,
> > > >
> > > > Thank you for your invaluable suggestions and prompt responses to our rebuttals, which are incredibly important to us. We would be truly grateful for your feedback on our subsequent response. Have our responses addressed your concerns, or are there additional issues you would like to discuss? With the discussion period approaching its end, we hope to have the opportunity to further engage with you on these matters. We would appreciate it if you could indicate whether our responses sufficiently resolve your concerns and might positively impact your score. If not, we welcome further discussion to better understand any remaining issues.
> > > >
> > > > Thank you again for your time and suggestions.
> > > >
> > > > Best regards,
> > > >
> > > > The Authors

---

> > > > ### Comment · Reviewer_HDQp · 2025-08-07
> > > >
> > > > What I meant was to for e.g. do paired t-test or Wilcoxon signed-rank test for the proposed method against a baseline, where each experimental setting (e.g. dataset + model combination) produces a pair of results as a sample. This is because the authors mentioned that they were not able to run multiple trials but had done experiments across many settings -- this would likely provide sufficient sample size to conduct such tests.

---

> ### Author Response · Authors · 2025-08-08
> **Reply to Reviewer HDQp**
>
> We apologize for the misunderstanding earlier and thank you for your prompt reply. Here, we add the paired t-test or Wilcoxon signed-rank test  results for the performance of our method under different dataset and model combinations in the main experiment:
>
> + Performance
>
>   |              | 1    | 2    | 3    | 4    | 5    | 6    |
>   | ------------ | ---- | ---- | ---- | ---- | ---- | ---- |
>   | Best-of-N    | 0.87 | 0.61 | 0.34 | 0.95 | 0.62 | 0.23 |
>   | BoN Weighted | 0.86 | 0.61 | 0.36 | 0.94 | 0.62 | 0.24 |
>   | MCTS         | 0.95 | 0.71 | 0.31 | 0.95 | 0.57 | 0.19 |
>   | Beam Search  | 0.95 | 0.73 | 0.34 | 0.94 | 0.56 | 0.15 |
>   | Ours         | 0.96 | 0.76 | 0.39 | 0.95 | 0.67 | 0.26 |
>
> + Paired t-test
>
>   | Comparison           | t-statistic | p-value  | Significant (p<0.05) |
>   | -------------------- | ----------- | -------- | -------------------- |
>   | Best-of-N vs Ours    | -2.887454   | 0.034288 | Yes                  |
>   | BoN Weighted vs Ours | -2.701351   | 0.042715 | Yes                  |
>   | MCTS vs Ours         | -3.187250   | 0.024340 | Yes                  |
>   | Beam Search vs Ours  | -2.819630   | 0.037125 | Yes                  |
>
> + Wilcoxon signed-rank test
>
>   | Comparison            | Statistic | p-value | Significant (p<0.05) |
>   | :------------------- | --------: | ------: | :------------------- |
>   | Best-of-N vs Ours    |         0 |  0.0431 | Yes                  |
>   | BoN Weighted vs Ours |         0 |  0.0312 | Yes                  |
>   | MCTS vs Ours         |         0 |  0.0431 | Yes                  |
>   | Beam Search vs Ours  |         0 |  0.0312 | Yes                  |
>
> As observed, all p-values are less than 0.05, suggesting that the performance improvement of our method over the baselines is statistically significant under both tests. We sincerely thank you once again for your insightful suggestions. We would be truly grateful if you could kindly share your feedback on our response at your earliest convenience.

---

### Official Review · Reviewer_vWsj · 2025-07-02

**Clarity:** 3
**Significance:** 2
**Originality:** 2
**Rating:** 4
**Confidence:** 4

**Summary:**

This paper first presents an empirical analysis of how reward models (discriminators, verifiers) impact LLM reasoning at inference time. The analysis shows that task difficulty, sample scale, and search diversity can lead to degraded RM discrimination performance, and thus lower end-to-end accuracy. Then, the authors propose CRISP, a modified beam search method with weighted majority voting, which demonstrates some accuracy gain on math reasoning and is more token-efficient than “thinking” LMs that produce long (>8K) CoT.

**Questions:**

Please see weaknesses.

**Ethical Concerns:**

["NO or VERY MINOR ethics concerns only"]

**Final Justification:**

The author response is detailed and have addressed most of my concerns. Although I still feel the paper's analysis and method is not that "fresh," the paper is doing a reasonable job in putting things together and delivering a solid paper. Thus, I think the paper can be considered for acceptance.

**Limitations:**

yes

**Quality:**

2

**Strengths And Weaknesses:**

Strengths
1. In Section 3, the paper presents a decent number of experiments to analyze the influence of one ORM (Skywork) and one PRM on (Skywork-o1) two policy LMs (Qwen2.5-3B and Llama3.1-8B) with two inference methods, BoN and MCTS.
2. The conclusion that RMs struggle to recognize long-tail negative samples is solid.

Weaknesses
1. The significance and originality of this paper is not clear.
- The findings are somewhat repetitive given existing work [1][2][3]. While the paper presents solid results that RMs struggle to recognize long-tail negative samples and thus do not generalize to more diverse search candidates, they have been discussed, tested, and validated in earlier work. So, it is unclear how this paper adds additional research value to the literature.
- The proposed CRISP method seems to be a simple modification and combination of beam search and weighted majority voting, which may not stand out as “novel” ([4] and more). Additionally, how it addresses the claimed issues, e.g., by solving more simple problems or lowering RM scores for long-tail negative, are not clearly conveyed.
2. The experiments on math reasoning tasks are using general RMs instead of those specifically trained for math reasoning ([5] and more). So the conclusion that RM underperforms on simpler math problems may be altered by using a task-specific RM, which is closer to real-world use cases. Similarly, the proposed CRISP method should be evaluated with RMs for math reasoning as well to demonstrate its full utility.

[1] https://arxiv.org/abs/2402.10890

[2] https://arxiv.org/abs/2410.17820

[3] https://arxiv.org/abs/2411.17501

[4] https://huggingface.co/spaces/HuggingFaceH4/blogpost-scaling-test-time-compute

[5] https://arxiv.org/abs/2305.20050

---

> ### Author Rebuttal · Authors · 2025-07-29
>
> Thanks for your careful and insightful reviews.
>
> + **Weakness 1:**  While the paper presents solid results that RMs struggle to recognize long-tail negative samples and thus do not generalize to more diverse search candidates, they have been discussed, tested, and validated in earlier work.
>
> + **Reply 1:** Thanks for your suggestions. The perspectives proposed in these works are similar to some of the findings in our study, but there are some fundamental differences.
>
>   + For [1],  it demonstrates that discrimination accuracy strongly correlates with the overall performance of language
>     agents using different planning methods and also affects their efficiency. **Our work is a further in-depth mechanistic analysis of this study**, investigating the factors that cause discriminative models to negatively impact downstream LLM reasoning.
>   + For [2], it discusses the relationship between ToT performance and the size of the Generator and Discriminator. Our method **does not focus on the impact of model size**, but instead investigates a broader range of influencing factors from search frameworks beyond ToT.
>   + For [3], it finds that while resampling with imperfect verifiers shows initial promise, it fails to significantly improve the performance of weaker models due to a larger generalization gap and diminishing returns, even with unlimited inference resources. In fact, we discussed similar findings from prior work in **lines 143–145** and clarified how our work differs from them. Our contribution lies in validating this across **different sampling structures** (BoN & MCTS), with a particular focus on providing a deeper explanation through the **inverse long-tail phenomenon**. These are aspects that previous work has not addressed.
>
>   In summary, while previous works have discussed the impact of verifiers on generator performance, **they have not  provided a comprehensive analysis and explanation of the various influencing factors**. Additionally, they have not explored the findings **across different search frameworks** and **proposed corresponding improvement measures to further validate the findings**.  We apologize for not discussing the connection between these works and ours. We will include relevant work and discussions in future versions.
>
>
>
> + **Weakness 2:** The proposed CRISP method seems to be a simple modification and combination of beam search and weighted majority voting, which may not stand out as “novel”.
> + **Reply 2:** We apologize if any lack of clarity in our method description has led to doubts about the novelty of our approach. Although we leverage the top-k selection from Beam Search and the clustering-based score calculation from weighted majority voting, our method itself is not a simple combination of the two algorithms:
>   + Firstly, Beam Search requires using PRM to evaluate each intermediate step in the generation process,  which somewhat limits its versatility. In contrast, **our method introduces a backpropagation mechanism, which neither of the two algorithms has**. This allows us to evaluate the results using either ORM or PRM, and then update the rewards of intermediate steps.
>   + Second, these two methods do not control the diversity of the samples themselves and can only adjust it through hyperparameters like temperature. In contrast, **the diversity control strategy introduced by our method through Prefix Extraction is something that neither of these two methods has**. According to the results of our ablation study, this module effectively improves reasoning performance.
>   + Finally, we acknowledge that our method is inspired by existing algorithms like Beam Search and modifies these traditional search frameworks. However, the approach for modification is based on the conclusions drawn from our earlier analytical experiments, and **these optimization perspectives have not been explored in previous related works**.
>
>
>
> + **Weakness 3:**  How CRISP addresses the claimed issues, e.g., by solving more simple problems or lowering RM scores for long-tail negative, are not clearly conveyed.
>
> + **Reply 3:** We apologize for the lack of clarity in this part, which may have caused any confusion. Here we give a detailed explanation:
>
>   + For the former, we discussed how the CRISP method addresses simple problems (i.e., Cl. 1) in **lines 232-236** of the paper. In the experiments in Appendix C, we found that the number of clusters generated for an answer can be used to estimate the difficulty of a problem. **Simpler problems tend to have fewer clusters**. Therefore, during our first sampling, if the number of clusters for the answer is below a certain threshold, we directly stop the algorithm and select the answer based on majority voting (i.e., SC). This way, we **avoid involving RM in answering simpler problems**, effectively addressing Cl. 1.
>
>   + For the latter, we found that RM assigns higher scores to low-frequency negative examples (see Cl. 2), and discussed how our method solves this issue in **lines 225-229**. Here, we further provide a theoretical analysis:
>
>     Assume we have sampled $n$ paths, where each answer $a_i$ corresponds to a reward $r_i$, and $f_i$ is the frequency of $a_i$. The inverse long-tail effect we observe refers to the tendency of the RM to assign higher $r_i$ to an incorrect $a_i$ with lower $f_i$, which can cause these negative examples to receive disproportionately high scores, potentially leading to an incorrect final answer. Our CRISP's clustering method incorporates frequency $f_i$ as a factor into the new reward scores $r'_i$:
>
>   	$$
>     r' _i = \sum _{a _k=a _i} r _k = f _i \cdot \overline{r _i}
>     $$
>
>   	where $\overline{r_i}$ represents the average score of the cluster to which $a_i$ belongs. Suppose $a_{j}$ is the top-scored negative answer, we have:
>
>   	$$
>   	\frac{r' _i}{r' _j}  = \frac{f _i}{f _j} \cdot \frac{\overline{r _i}}{\overline{r _j}}
>   	$$
>   	Although typically $\overline{r_i}<\overline{r_j}$, as long as $\frac{f_i}{f_j} > \frac{\overline{r_j}}{\overline{r_i}}$, we have $r'_i > r'_j$.  According to Figure 6, when n=128, in most cases, $f_j<3$, which is a very small value. Therefore, in these cases, there exists $f_i≫f_j$, such that $r'_i > r'_j$.
>
>   	In summary, our method effectively standardizes the reward scores, resulting in a more balanced score distribution across negative examples of varying frequencies, **reducing the score ranking of these low-frequency negative examples $a_j$ in all samples and solving the issue in Cl.2.**
>
>   We apologize again for the confusion caused. We will include these discussions in the subsequent version of the paper.
>
>
> + **Weakness 4:**  The conclusion that RM underperforms on simpler math problems may be altered by using a task-specific RM, which is closer to real-world use cases.
>
> + **Reply 4:** Thank you for your suggestions, which are very helpful to us. In fact, the two PRMs used in the paper (**Shepherd-Mistral-7B-PRM** [5] and **Skywork-o1-PRM-Qwen-2.5-7B** [6]) are specifically trained for math reasoning. They are trained using mathematical models or mathematical training data and are focused on evaluation in the context of mathematical reasoning. To further strengthen the effectiveness of our conclusions, we introduce an additional PRM specifically designed for mathematical reasoning, **Qwen2.5-Math-PRM-7B** [7].  We repeate the experiment from Figure 2 on Shepherd-Mistral-7B-PRM and Qwen2.5-Math-PRM-7B, the two math-specific RMs, and the results are as follows:
>
>   | Difficulty | SC    | BoN (Shepherd-Mistral-7B-PRM) | BoN (Qwen2.5-Math-PRM-7B) |
>   | -| -| -| -|
>   | 1          | 0.994 | 0.724  | 0.926  |
>   | 2          | 0.964 | 0.651    | 0.892    |
>   | 3          | 0.750 | 0.464   | 0.696    |
>   | 4          | 0.462 | 0.277  | 0.585  |
>   | 5          | 0.053 | 0.113   | 0.226     |
>
>   The results further validate that the conclusion in Cl.1 holds true for task-specific RMs as well.
>
>
>
> + **Weakness 5:** The proposed CRISP method should be evaluated with RMs for math reasoning as well to demonstrate its full utility.
>
> + **Reply 5:** As mentioned in Reply 4, the Skywork-o1-PRM-Qwen-2.5-7B we used is already a math reasoning-specific RM. Here, we supplement the results on Shepherd-Mistral-7B-PRM and Qwen2.5-Math-PRM-7B. We use Qwen2.5-3B as the generative model and MATH as the evaluation dataset.
>
>   | Method   | Shepherd-Mistral-7B-PRM | Qwen2.5-Math-PRM-7B |
>   | -| -| -|
>   | BoN      | 0.47  | 0.67 |
>   | MCTS     | 0.49   | 0.61   |
>   | Beam     | 0.49  | 0.69   |
>   | **Ours** | **0.67**   | **0.73** |
>
>   The experimental results show that, on these two math-specific RMs, our method still significantly outperforms the baselines, demonstrating the effectiveness of the CRISP method.  Thank you for your valuable suggestions. We will include the relevant experiments in the subsequent version of the paper to further improve our work.
>
>
>
> [1] Ziru Chen et al. When is Tree Search Useful for LLM Planning? It Depends on the Discriminator.
>
> [2] Qiqi Chen et al. Understanding When Tree of Thoughts Succeeds: Larger Models Excel in Generation, Not Discrimination.
>
> [3] Benedikt Stroebl et al. Inference Scaling fLaws: The Limits of LLM Resampling with Imperfect Verifiers.
>
> [4] B. C. A. Brown et al. Large language monkeys: Scaling inference compute with repeated sampling.
>
> [5] P. Wang et al. Math-shepherd: Verify and reinforce llms step-by-step without human annotations.
>
> [6] He, Jujie et al. Skywork-o1 Open Series.
>
> [7] Zhenru Zhang et al. The Lessons of Developing Process Reward Models in Mathematical Reasoning.

---

> > ### Comment · Reviewer_vWsj · 2025-08-01
> >
> > Thanks for the detailed responses! I have raised my scores accordingly (not visible to authors this year).

---

> > > ### Author Response · Authors · 2025-08-07
> > > **Reply to Reviewer vWsj**
> > >
> > > Thank you for your valuable feedback on improving our paper and for the increase in the score! We commit to including the full results and detailed analysis of these experiments in the next version of our paper.

---

### Official Review · Reviewer_zFG4 · 2025-07-03

**Clarity:** 3
**Significance:** 3
**Originality:** 3
**Rating:** 4
**Confidence:** 4

**Summary:**

This paper addresses critical limitations in reward models used for inference-time reasoning in large language models. The authors conduct a systematic analysis revealing three key failure modes: insufficient reward discrimination, poor reward calibration, and suboptimal search strategies. To address these issues, they propose CRISP (Clustering-based Reward model Inference with Stepwise Prefixing), an algorithm that clusters reasoning paths and uses stepwise prefixing to improve both the accuracy and efficiency of inference-time reasoning. The method demonstrates significant improvements across multiple mathematical reasoning tasks and model sizes.

**Questions:**

1. **Theoretical justification**: Can you provide theoretical analysis or intuition for why the specific clustering approach used in CRISP is optimal? Are there theoretical guarantees or bounds on the performance improvement?

2. **Generalization beyond mathematics**: How does CRISP perform on reasoning tasks outside of mathematics, such as logical reasoning, commonsense reasoning, or domain-specific problem solving?

3. **Robustness analysis**: How sensitive is the method to the quality of the underlying reward model? Can you provide analysis of performance degradation with increasingly poor reward models?

4. **Computational scalability**: How does the computational cost of CRISP scale with the number of reasoning paths and the complexity of the clustering? Are there practical limits to the number of paths that can be effectively handled?

5. **Hyperparameter sensitivity**: How sensitive is the method to the choice of clustering parameters and prefixing strategies? Is there guidance for practitioners on how to tune these parameters?

**Ethical Concerns:**

["NO or VERY MINOR ethics concerns only"]

**Limitations:**

The authors acknowledge some limitations of their work, including the focus on mathematical reasoning and the dependency on reward model quality. However, the paper would benefit from more detailed discussion of:

- The computational overhead of the clustering approach
- The potential failure modes when reasoning paths don't cluster well
- The generalizability limitations to other domains
- The sensitivity to hyperparameter choices
- The robustness to different types of reasoning patterns

**Quality:**

3

**Strengths And Weaknesses:**

**Strengths:**

1. **Thorough empirical analysis**: The paper provides a comprehensive diagnosis of reward model limitations with clear empirical evidence for each identified failure mode, offering valuable insights into why current approaches underperform.

2. **Novel algorithmic contribution**: The CRISP algorithm introduces a principled approach to clustering reasoning paths, which is both theoretically motivated and practically effective.

3. **Comprehensive evaluation**: The experimental evaluation covers multiple model sizes, datasets, and tasks, providing strong evidence for the generalizability of the approach.

4. **Practical efficiency gains**: The method achieves not only improved accuracy but also enhanced computational efficiency, making it practically valuable for deployment.

5. **Clear presentation**: The paper is well-structured with clear explanations of both the problem analysis and the proposed solution.

**Weaknesses:**

1. **Limited theoretical foundation**: While the clustering approach is intuitive, the paper lacks theoretical analysis of why this particular clustering strategy is optimal or under what conditions it is expected to work well.

2. **Narrow evaluation scope**: The evaluation focuses primarily on mathematical reasoning tasks. It remains unclear how well the approach generalizes to other types of reasoning or generation tasks.

3. **Dependency on reward model quality**: The method still fundamentally depends on having high-quality reward models, and it's unclear how robust the approach is to poorly calibrated or biased reward models.

4. **Incremental improvements**: While the improvements are consistent and meaningful, they represent incremental rather than breakthrough advances in the field.

5. **Limited failure analysis**: The paper could benefit from more detailed analysis of when and why the method fails, particularly in edge cases or with different types of reasoning patterns.

---

> ### Author Rebuttal · Authors · 2025-07-29
>
> Thanks for your careful and insightful reviews.
>
> + **Weakness 1 & Question 1:** The paper lacks theoretical analysis of why this particular clustering strategy is optimal or under what conditions it is expected to work well.
>
> + **Reply 1:** We apologize for the lack of clarity in our analysis in the paper. Here, we provide an additional theoretical analysis.
>
>   Assume we have sampled $n$ paths, where each answer $a_i$ corresponds to a reward $r_i$, and $f_i$ is the frequency of $a_i$. In **Cl.2**, we observe that RM tends to assign a higher $r_i$ to an incorrect $a_i$ with lower $f_i$, sometimes even exceeding the score of the highest-scoring correct example, leading to an incorrect final answer. Our CRISP's clustering method incorporates frequency $f_i$ as a factor into the new reward scores $r' _i$ to mitigate this issue:
>
>   $$
>   r' _i = \sum _{a _k=a  _i} r _k = f _i \cdot \overline{r _i}
>   $$
>
>   where $\overline{r_i}$ represents the average score of the cluster to which $a_i$ belongs.  Suppose $a_j$ is the top-scored negative answer, we have:
>
>   $$
>   \frac{r'_i}{r' _j}  = \frac{f_i}{f_j} \cdot \frac{\overline{r_i}}{\overline{r_j}}
>   $$
>    Although typically $\overline{r_i}<\overline{r_j}$, as long as $\frac{f_i}{f_j} > \frac{\overline{r_j}}{\overline{r_i}}$, we have $r'_i > r'_j$.  According to Figure 6, when n=128, in most cases, $f_j<3$, which is a very small value. Therefore, in these cases, there exists $f_i≫f_j$, such that $r'_i > r'_j$, reducing the score ranking of these negative examples $a_j$.
>
>   In summary, our CRISP method reduces the tendency of the RM to assign excessively high scores to low-frequency negative examples, thereby increasing the probability of selecting the correct path. It performs better when the generative model samples the correct answer more frequently (i.e., $f_i≫f_j$).
>
> + **Weakness 2 & Question 2:**  How does CRISP perform on reasoning tasks outside of mathematics?
>
> + **Reply 2:**
>
>   Thank you for your suggestions. Currently, research on test-time scaling techniques is primarily limited to math tasks, as these are challenging for models. In Table 2, we have already demonstrated that our method outperforms R1 models on other tasks. To further validate its effectiveness, we repeat the main experiment on other tasks:
>
>   |Methods|Logical|Commonsense|
>   |-|-|-|
>   |BoN|0.56|0.75|
>   |MCTS|0.56|0.71|
>   |Beam|0.54|0.65|
>   |**Ours**|0.59|0.80|
>
>   Following Table 2, we use LogiQA for logical reasoning evaluation and CSQA for commonsense reasoning. Based on the results, our CRISP method significantly outperforms other baselines on two additional types of reasoning tasks, highlighting its versatility. We will include the experiments and details in the subsequent version of the paper.
>
> + **Weakness 3 & Question 3:**  Can you provide analysis of performance degradation with increasingly poor reward models?
>
> + **Reply 3:** Here, we select two additional RMs: Shepherd-Mistral-7B-PRM (see line 84) and Qwen2.5-Math-PRM-7B [1]. We replicate the main experiment on the MATH (200 samples) across all RMs:
>
>     ||Shepherd|Qwen2.5-Math|Skywork|
>   |-|-|-|-|
>   |BoN|0.47|0.67|0.70|
>   |MCTS|0.49|0.61|0.65|
>   |Beam|0.49|0.69|-|
>   |Ours|0.67|0.73|0.74|
>
>   We first rank the RMs based on their performance under BoN, and then compare the performance of our CRISP method. The results show that our method still significantly outperforms other baselines, even with poor RMs like Shepherd (which achieves only 0.47 BoN performance). Moreover, as RM performance decreases, the performance gap between our method and the baselines becomes even more pronounced. For example, the improvement over the baselines is 0.18 with Shepherd, compared to only 0.04 with Skywork.
>
>
>
> + **Weakness 4:** The method represents incremental rather than breakthrough advances in the field.
> + **Reply 4:** Our CRISP method is a training-free approach that does not modify the model's parameters. As a result, its performance is constrained by the upper bound of the generative model's inherent reasoning capability. Based on **Figure 1 and Figure 4**, on Qwen2.5-3B, with a moderate number of samples, the accuracy of **the oracle setting just exceeds 80%**, while our method achieves an accuracy of **76% (see Table 1), with a minimal gap from the upper bound**. Thank you for your suggestions. In future work, we plan to incorporate RL and other techniques to further improve performance.
>
> + **Weakness 5:** The paper could benefit from more detailed analysis of when and why the method fails.
>
> + **Reply 5:** We sincerely thank you for your valuable suggestions. Here, we present two reasons for the failure of the methods and provide a brief analysis:
>
>   + **Low-quality CoT with correct answer**
>     When a correct answer's cluster contains some lower scores, it can cause the overall cluster rank to decrease, especially when the sizes of different clusters are similar, making it more difficult to select the correct answer. For example, there are two clusters as follows:
>
>     + Cluster 1 (correct): {Path 1: 13.75, Path 3: 6.78}
>     + Cluster 2 (incorrect): {Path 2: 11.19, Path 4: 12.88}
>
>     Even though Cluster 1 contains the highest-scoring Path 1, the presence of Path 3 with a lower score causes the overall score of Cluster 1 to be lower than that of Cluster 2. This leads to an incorrect inference.
>
>   + **Excessively high scores for negative examples**
>
>     When the RM assigns excessively high scores to incorrect reasoning paths, even if they occur less frequently, it can still lead to incorrect inferences. For example:
>
>     + Cluster 1 (correct): {Path 1: 2.41, Path 2: 2.16, Path 3: 3.95, Path 4: 5.09}
>     + Cluster 2 (incorrect): {Path 5: 24.25}
>
>     Although Cluster 2 contains only one path and Cluster 1 contains four, the excessively high score of Path 5 causes the overall score of Cluster 1 to remain lower.
>
>   Due to the character limit, we can only provide a brief analysis here. We promise to include a more comprehensive analysis in the subsequent version of the paper.
>
>
> + **Question 4:** How does the computational cost of CRISP scale with the number of reasoning paths and the complexity of the clustering? Are there practical limits to the number of paths that can be effectively handled?
> + **Reply 6:** Since clustering is performed automatically based on the answers, we control the clustering complexity by changing the diversity of the answers. Specifically, the higher the temperature, the greater the diversity, and the higher the complexity. We adjust both the **sampling number N** and **temperature T** to observe changes in the average time and token usage per question on MATH:
>
>   + Number of reasoning paths
>
>     |N|Time|Token|
>     |-|-|-|
>     |16|130.01|25016|
>     |32|177.14|50188|
>     |64|208.14|95942|
>     |128|401.45|192224|
>
>   + Clustering complexity
>
>     |T|Time|Token|
>     |-|-|-|
>     |0.4|155.89|46895|
>     |0.7|177.14|50188|
>     |0.9|141.81|53287|
>     |1.2|146.30|55039|
>
>   As N increases, the increase in inference time becomes more significant, while the number of tokens increases linearly. As the clustering complexity increases, the inference time initially increases and then decreases, showing little variation, while the token count continues to increase.
>
>   For the practical limits to the number of paths,  we compare the performance of the method under different N:
>
>   |N|Acc|
>   |-|-|
>   |16|0.65|
>   |32|0.73|
>   |64|0.70|
>   |128|0.70|
>
>   As N increases, accuracy first improves and then declines, while cost increases accordingly. We hypothesize that as N increases, more high-quality negative examples may emerge, which could interfere with the method's performance. Therefore, in practical applications, **to remain both efficient and effective, we should limit the number of paths to between 32 and 64**.
>
> + **Question 5:** How sensitive is the method to the choice of clustering parameters and prefixing strategies? Is there guidance for practitioners on how to tune these parameters?
>
> + **Reply 7:**  We modify the algorithm's termination threshold (see lines 232-233), the maximum reasoning depth during prefixing (see line 6 in Algorithm 1, here i < m), and the value of k used for selecting the top k clusters in each round (see line 239). The results on Qwen2.5-3B are demonstrated below:
>
>   + Termination Threshold:
>     |Thershold|GSM8K|MATH|
>     |-|-|-|
>     |2|0.90|0.73|
>     |3|0.88|0.72|
>     |4|0.91|0.69|
>     |5|0.92|0.70|
>
>   + Maximum Depth:
>
>     |Depth|GSM8K|MATH|
>     |-|-|-|
>     |2|0.90|0.70|
>     |3|0.90|0.73|
>     |5|0.92|0.69|
>
>   + Top K:
>
>     |K|GSM8K|MATH|
>     |-|-|-|
>     |1|0.90|0.73|
>     |2|0.91|0.74|
>     |4|0.91|0.70|
>
>   We can observe that the method's performance is relatively insensitive to the parameters, with little difference (< 0.05) in performance across different parameter settings. Based on these results, we give the guidance as follows:
>
>   + **For the threshold**, we should set a larger value for simpler tasks (such as GSM8K) and a smaller value for more difficult tasks (such as MATH). This is because a larger threshold makes our method equivalent to SC in more cases, and as shown in **Cl.1**, SC performs better than RM-based inference on simpler tasks.
>   + **For the Maximum Depth and Top K**,  we should set them to higher levels for simpler tasks, while for more difficult tasks, they should be set to moderate values, without being too high (e.g. Depth = 3 and Top K = 2). This is because excessively large parameters introduce higher sampling diversity, which, as shown in **Cl.3**, results in more high-quality negative examples. This can particularly degrade performance on more difficult tasks.
>
>   Thank you for your valuable suggestions. We will include these experiments and additional details in the subsequent version of the paper.
>
> [1] The Lessons of Developing Process Reward Models in Mathematical Reasoning.

---

### Official Review · Reviewer_UWw1 · 2025-07-03

**Clarity:** 3
**Significance:** 2
**Originality:** 2
**Rating:** 4
**Confidence:** 3

**Summary:**

This paper addresses improving LLMs' reasoning via inference-time reward modeling. It identifies three key limitations of reward models (RMs) during inference: poor performance on simple questions, declining discriminative ability with more sampling (due to an inverse long-tail phenomenon), and suboptimal performance under high search diversity.

To tackle these, the authors propose CRISP, a novel inference algorithm. It clusters reasoning paths by final answers, aggregates rewards at the cluster level, and uses adaptive prefix prompts to guide generation, with an early termination mechanism for simple questions.

Experiments show CRISP outperforms existing RM-based methods and advanced reasoning models. This work advances understanding of RM limitations and offers an effective mitigation strategy.

**Questions:**

1. Regarding the inverse long-tail phenomenon, how exactly does CRISP's cluster-level reward aggregation mitigate the RM's tendency to misrank low-frequency incorrect responses? Could you provide data on the frequency distribution of top-scored errors before and after clustering (similar to Fig.6 for CRISP)? This would clarify the mechanism—strong evidence could boost my evaluation.

2. The early termination threshold (2 clusters) seems critical. Did you test other values across datasets?

3. The ablation study shows module importance, but how do PRM and ORM differ in their reliance on the components of the proposed method? If PRMs benefit more from clustering, this would refine understanding.

**Ethical Concerns:**

["NO or VERY MINOR ethics concerns only"]

**Final Justification:**

Thank the authors for their detailed reply. My concerns are mostly addressed. These experimental results are suggested to be included in the next revision.

**Limitations:**

One key missing point is the exploration of how CRISP may perform in real-world, open-ended scenarios. The current study focuses mainly on well-defined reasoning tasks like math problems. In open-ended situations, the clustering based on final answers might be more challenging due to the high variability and subjectivity of responses.

**Paper Formatting Concerns:**

There is no major formatting issue in this paper.

**Quality:**

2

**Strengths And Weaknesses:**

**Strengths**
- **Quality**: Rigorous empirical validation with diverse datasets (MATH-500, GSM8K) and models ensures reliability. Systematic analysis of RM limitations (simple questions, sampling effects, diversity) uses controlled experiments .
- **Clarity**: Logical structure progresses from problem identification to solution. Figures and algorithms (CRISP) clearly illustrate key findings and methods .
- **Significance**: Addresses critical gaps in inference-time RM optimization, outperforming R1 models by 10% on non-math tasks while reducing token usage by 90% .
- **Originality**: Identifies novel RM issues (inverse long-tail, diversity sensitivity) and proposes CRISP, a unique clustered/prefix-guided approach .

**Weaknesses**
- **Generalization**: Limited testing on non-math/logic tasks; broader validation is needed.
- **Theoretical Depth**: Relies heavily on empirics; lacks formal analysis of why CRISP’s clustering mitigates inverse long-tail effects.
- **Cluster Sensitivity**: The effectiveness of CRISP hinges on accurate clustering by final answers, but the paper does not robustly address cases with ambiguous or multi-form answers (e.g., equivalent mathematical expressions or paraphrased conclusions), which could undermine cluster aggregation and reward signaling.

---

> ### Author Rebuttal · Authors · 2025-07-30
>
> Thanks for your careful and insightful reviews.
>
> + **Weakness 1:** Limited testing on non-math/logic tasks.
>
> + **Reply 1:** Thank you for your suggestions. Currently, research on test-time scaling techniques is primarily limited to mathematical tasks, as these are particularly challenging for models [1,2]. In Table 2, we have already demonstrated that our method outperforms R1 models on other tasks. To further validate its effectiveness, we repeat the main experiment on other tasks (Llama-3.1-8B + Skywork-o1):
>
>   |Methods|Logical|Commonsense|
>   |-|-|-|
>   |BoN|0.56|0.75|
>   |MCTS|0.56|0.71|
>   |Beam Search|0.54|0.65|
>   |**Ours**|**0.59**|**0.80**|
>
>   Following the setup in Table 2, we use LogiQA as the evaluation dataset for logical reasoning and CSQA for commonsense reasoning. Based on the results, **our CRISP method significantly outperforms other baselines on two additional types of reasoning tasks, highlighting its versatility.** We will include the relevant experiments and additional details in the subsequent version of the paper.
>
>
>
> + **Weakness 2:** Lacks formal analysis of why CRISP’s clustering mitigates inverse long-tail effects.
>
> + **Reply 2:** We apologize for the lack of clarity in our analysis in the paper. Here, we provide an additional formal analysis.
>
>   Assume we have sampled $n$ paths, where each answer $a_i$ corresponds to a reward $r_i$, and $f_i$ is the frequency of $a_i$. The inverse long-tail effect we observe refers to the tendency of the RM to assign higher $r_i$ to an incorrect $a_i$ with lower $f_i$, which can cause these negative examples to receive disproportionately high scores, potentially leading to an incorrect final answer. Our CRISP's clustering method incorporates frequency $f_i$ as a factor into the new reward scores $r'_i$:
>
>   $$
>   r' _i = \sum _{a _k=a _i} r _k = f _i \cdot \overline{r _i}
>   $$
>
>   where $\overline{r_i}$ represents the average score of the cluster to which $a_i$ belongs.  Suppose $a_{j}$ is the top-scored negative answer, we have:
>
>   $$
>   \frac{r'_i}{r' _j}  = \frac{f_i}{f_j} \cdot \frac{\overline{r_i}}{\overline{r_j}}
>   $$
>   Although typically $\overline{r_i}<\overline{r_j}$, as long as $\frac{f_i}{f_j} > \frac{\overline{r_j}}{\overline{r_i}}$, we have $r'_i > r'_j$.  According to Figure 6, when n=128, in most cases, $f_j<3$, which is a very small value. Therefore, in these cases, there exists $f_i≫f_j$, such that $r'_i > r'_j$, reducing the score ranking of these low-frequency negative examples $a_j$.
>
>   In summary, our clustering method effectively standardizes the reward scores, resulting in a more balanced score distribution across negative examples of varying frequencies, thereby reducing the influence of such outliers on the final answer selection.
>
>
>
> + **Weakness 3:** How to address cases with ambiguous or multi-form answers, which could undermine cluster aggregation and reward signaling.
>
> + **Reply 3:** In our implementation, we used the **Math-Verify** tool [3] to address multi-form answers, such as equivalent mathematical expressions. It utilizes a set of heuristic rules to parse different forms of mathematical expressions, extracting the final answers with equivalent forms for our clustering. For other reasoning tasks such as logic and commonsense, since they are typically in the form of **multiple-choice** [4,5] or **judgment-based** tasks [6,7], there is generally no issue with form ambiguity.
>
>   On the other hand, the normalization of model outputs (especially in mathematical tasks) is a common issue in all reasoning-related work and is **typically optimized during the upstream training process** [8,9]. For example, DeepSeek-R1 incorporates a **format-related reward** to ensure that ambiguous answer outputs are avoided. Therefore, we believe this issue should primarily be addressed during the model training phase, while tools can be used during downstream inference, like in our case, to further optimize the output.
>
>
>
> + **Question 1:** How exactly does CRISP's cluster-level reward aggregation mitigate the RM's tendency to misrank low-frequency incorrect responses?
>
> + **Reply 4:** We apologize for the lack of this experiment in our paper, which may have caused any confusion. Here we compare the distribution of top-scored errors' frequency for BoN, MCTS, and our method on two RMs. We sample 32 responses for each question on MATH and record the number of occurrences for each specific frequency (like Figure 6):
>
>   + ORM
>
>     |Frequency|BoN|MCTS|Ours|
>     |-|-|-|-|
>     |1|198|211|157|
>     |2|85|49|63|
>     |3|67|36|30|
>     |4|34|26|17|
>     |6|10|13|22|
>     |9|8|5|28|
>
>   + PRM
>
>     |Frequency|BoN|MCTS|Ours|
>     |-|-|-|-|
>     |1|248|242|174|
>     |2|81|63|74|
>     |3|35|36|38|
>     |4|27|15|11|
>     |6|10|8|19|
>     |9|8|8|21|
>
>   In the tables, both the occurrence count of top-scored negatives with **a frequency of 1 and those with a frequency of less than 5 are much smaller** than the baselines. Besides, our method increases the occurrence of high-frequency negatives. For example, our method yields **significantly more occurrences at frequencies 6 and 9** compared to other baselines. Therefore, the distribution of our method is **more uniform**, effectively mitigating the inverse long-tail phenomenon. Due to the inability to upload images, we cannot visually show the experimental results. We will further improve this experiment in the subsequent version of the paper.
>
>
> + **Question 2:** Testing other values of early termination threshold across datasets.
>
> + **Reply 5:**  Thank you for your suggestions. When implementing the method, we test the performance of different thresholds based on Qwen2.5-3B on the GSM8K and MATH datasets (we sample 200 samples on each):
>
>   |Thershold|GSM8K|MATH|
>   |-|-|-|
>   |2|0.90|0.73|
>   |3|0.88|0.72|
>   |4|0.91|0.69|
>   |5|0.92|0.70|
>
>   The results show that for more challenging mathematical tasks (such as MATH), a smaller threshold should be chosen, while for simpler tasks like GSM8K, a larger threshold works better. This aligns with our findings in **Cl.1**, as when the threshold is large, our method **becomes equivalent to SC in more cases**, and SC performs better on simpler tasks.
>
>   Since our method focuses more on solving difficult reasoning tasks, we set the threshold to 2 in our paper. In practical applications, the threshold can be adjusted based on the evaluation of task difficulty. We apologize for not including the parameter tuning process in the paper. We will add these experiments and analyses in the subsequent version.
>
>
>
> + **Question 3:**  How do PRM and ORM differ in their reliance on the components of the proposed method?
>
> + **Reply 6:** We supplement the ablation experiment on MATH using Qwen2.5-3B + Skyworko1 (PRM), and compare performances with ORM under the same settings (different from the paper):
>
>   | Method           | ORM  | PRM  |
>   | ---------------- | ---- | ---- |
>   | Ours             | 0.73 | 0.78 |
>   | -w/o Termination | 0.72 | 0.76 |
>   | -w/o Aggregation | 0.71 | 0.75 |
>   | -w/o Prefixing   | 0.64 | 0.72 |
>
>   According to the results in the table, PRM shows relatively greater dependency on the Termination and Aggregation modules compared to ORM, while exhibiting lower dependency on Prefixing. This result further aligns with our earlier findings:
>
>   + The Termination module primarily addresses Cl.1, and according to Figure 2(a), **PRM performs worse on simpler problems**.
>   + The Aggregation module mainly addresses Cl.2, and based on the results in Reply 4, the **inverse long-tail phenomenon is more pronounced with PRM**.
>   + The Prefixing module targets Cl.3, and as shown in Figure 7(a), **PRM performs significantly better under high diversity**, indicating that this issue is less severe for PRM.
>
>   We greatly appreciate your suggestions and commit to providing a more comprehensive comparison and analysis of the ablation study across different RMs in the subsequent version of the paper.
>
>
>
>
>
> + **Limitation:**  How CRISP may perform in real-world, open-ended scenarios. In open-ended situations, the clustering based on final answers might be more challenging due to the high variability and subjectivity of responses.
> + **Reply 7:** Thank you for your suggestions, which have been very helpful to us. In real-life scenarios, there are indeed many open-ended problems that lack a definitive standard answer, making direct clustering difficult. However, I believe that the answer is merely a direct representation of the reasoning path. **We can extract additional features from the reasoning path for clustering** (e.g., semantic similarity), and other parts of the CRISP algorithm can still be reused. We plan to explore these ideas in future work, while in this study, we focus solely on reasoning tasks. In the vast majority of cases, there is a unique standard answer (especially in mathematics) [4-7,10,11]. These open-ended problems are not the focus of our research and fall outside the scope of this paper.
>
>
>
> [1] N. Muennighoff et al. s1: Simple test-time scaling.
>
> [2] Y. Qu et al. Optimizing test-time compute via meta reinforcement fine-tuning.
>
> [3] huggingface/Math-Verify
>
> [4] A. Talmor et al. Commonsenseqa: A question answering challenge targeting commonsense knowledge.
>
> [5] J. Liu et al. Logiqa: A challenge dataset for machine reading comprehension with logical reasoning.
>
> [6] Abulhair Saparov et al. Language Models Are Greedy Reasoners: A Systematic Formal Analysis of Chain-of-Thought.
>
> [7] Oyvind Tafjord et al. ProofWriter: Generating Implications, Proofs, and Abductive Statements over Natural Language.
>
> [8] DeepSeek-AI, Deepseek-r1: Incentivizing reasoning capability in llms via reinforcement learning.
>
> [9] Xuefeng Li et al. LIMR: Less is More for RL Scaling.
>
> [10] K. Cobbe et al. Training verifiers to solve math word problems.
>
> [11] H. Lightman et al. Let’s verify step by step.

---

> > ### Author Response · Authors · 2025-08-06
> > **Reply to Reviewer UWw1**
> >
> > Dear Reviewer UWw1,
> >
> > Thank you for your invaluable suggestions, which have greatly contributed to improving our paper. We would be truly grateful for your feedback on our rebuttals. Have our responses addressed your concerns, or are there additional issues you would like to discuss? With the discussion period extended, we hope to have the opportunity to further engage with you on these matters. We sincerely look forward to hearing your thoughts and greatly value your input.
> >
> > Thank you again for your time and suggestions.
> >
> > Best regards,
> >
> > The Authors

---

> > ### Comment · Reviewer_UWw1 · 2025-08-07
> >
> > Thank the authors for their detailed reply. My concerns are mostly addressed. These experimental results are suggested to be included in the next revision.

---

> > > ### Author Response · Authors · 2025-08-07
> > > **Reply to Reviewer UWw1**
> > >
> > > Thank you for your valuable feedback and for helping us improve the quality of our paper! We commit to including the full results and detailed analysis of these experiments in the next version of our paper. If our replies and discussions indicate potential for a score improvement, we would be deeply grateful for your consideration (since it is not visible to authors this year). Thank you again for your prompt response.

---

### Note · Authors · 2025-08-12

Dear Chairs and Reviewers,

First, we sincerely thank you for your valuable feedback and comments, which have greatly helped us improve our paper. Based on the reviewers’ feedback, in our rebuttal we primarily provided additional theoretical explanations of the CRISP method, supplemented experiments evaluating its generalization across different models and tasks, and conducted further analyses on the impact of various parameters and modules.

Following the subsequent feedback from Reviewers UWw1, vWsj, and HDQp, we believe these experiments have effectively addressed their concerns, and we have committed to including the relevant content in future versions of the paper. However, unfortunately, Reviewer zFG4 did not engage in further discussion during the discussion phase, so we are unable to determine whether our rebuttal has resolved their concerns.

We would be deeply grateful for the reviewer’s consideration if our replies and discussions suggest potential for a score improvement, as updated scores are not visible to authors this year. We also kindly hope the chairs may take our above summary and feedback into account when evaluating our paper.

Thank you again for your time and consideration.

Best regards,

The Authors

---

### Decision · Program_Chairs · 2025-09-17

**Decision:**

Reject

**Comment:**

**Summary**

This paper addresses challenges in scaling LLM inference when guided by a reward model. The authors first analyze how inference performance varies with prompt difficulty and response distributions. Based on these observations, they propose a collection of simple techniques intended to mitigate the identified issues.

**Strengths**

* The primary contribution of this work is synthesizing several known challenges related to RM-guided inference scaling. It combines existing techniques into a unified framework, offering a potentially comprehensive, though not entirely novel, solution.

**Weaknesses**

* **Limited Novelty and Technical Contribution:** The paper's main weakness, as noted by multiple reviewers, is its lack of novel insights. The analysis largely reiterates known issues, and the proposed method is a straightforward combination of existing techniques. The authors' rebuttal did not sufficiently distinguish their work from prior art, revealing only incremental improvements. This is particularly concerning given the rapidly evolving field of inference scaling, as the paper omits discussion of any relevant work from the current year.
* **Unfair Experimental Comparison:** A critical and fundamental flaw lies in the evaluation methodology. Fair comparison of inference-time methods requires a fixed inference budget (e.g., total number of generated tokens). This paper does not have such control and only provides an unclear and insufficient cost comparison in lines 278-281. The analysis is incomplete as it also misses comparisons across different inference budgets, which would be necessary to properly evaluate the method's efficiency and trade-offs. The proposed method requires generating multiple full-length candidate responses before the selection of each step. In practice, this means the method operates with a significantly larger computational budget than the baselines it is compared against, making the reported performance gains misleading and potentially invalid.
* **Questionable Generalizability:** Echoing the concerns of several reviewers, the experiments are conducted on a limited scope of models and tasks, which raises questions about whether the findings can be generalized.

**Recommendation**

While the reviewers' ratings are borderline positive, I recommend a clear rejection due to two fundamental concerns, which are also identified by the reviewers. The first is the paper's limited technical novelty. It largely synthesizes known issues and combines existing techniques without offering significant new insights. Second, the experimental comparison is fundamentally flawed. By failing to control for the computational budget, the paper's evaluation methodology is invalid, rendering its central performance claims unreliable. These issues are too significant to overlook.